# Föhn-induced melting over Larsen C modulated by atmospheric river shape, direction and landfall location

Xun Zou [1] ✉, Penny M. Rowe[2], Irina V. Gorodetskaya[3], Andrew Orr[4], David H. Bromwich [5,6], Dan Lubin [1], Matthew A. Lazzara [7,8], Zhenhai Zhang[1], Kawzenuk Brian[1], Jonathan D. Wille[9], Jason M. Cordeira[1], Nicolaj Hansen [10], Jinxi Li [11], Pu Gan[11] & F. Martin Ralph[1]

Recent decades have seen record-high temperatures on the Antarctic Peninsula (AP) due to combined atmospheric rivers (ARs) and föhn warming. While ARs frequently enhance föhn, not all events cause surface warming over the entire Larsen C Ice Shelf (LCIS). Using high-resolution Polar WRF simulations, we examine the relationship between ARs and föhn over the AP during austral summers and identify four distinct AR shapes associated with föhn-induced surface warming over the LCIS: zonal-perpendicular, zonal-like, convex, and concave. Zonal-like ARs associated with coupled low-high-pressure systems and convex ARs linked to blocking highs produce strong föhn warming across the entire LCIS, primarily affecting its northern and southern sectors, respectively. In contrast, zonal-perpendicular and concave ARs generate moderate-to-weak warming, owing to either weaker AR intensity or AR curvature. Although downward shortwave radiation dominates surface warming, enhanced moisture suppresses its increase from föhn-induced cloud clearance while enhancing downward longwave radiation near mountain gaps. Sensible heat flux also contributes substantially along the mountain foothills. As ARs intensify under climate change, their interaction with föhn over the AP can critically influence the future stability of coastal ice shelves.

The Antarctic Peninsula (AP) has warmed by ~3 °C over the past 60 years, despite a brief cooling since the late 1990s[1,2]. Notably, the northern AP (north of 70°S) experienced a summer warming rate of +0.15 °C per decade from 1959 to 2021, accompanied by multiple extreme warming events[3]. Extreme warming events like this have intensified surface melting over AP ice shelves, contributing to past collapses[4,5] and threatening the last remaining eastern AP ice shelf—the Larsen C Ice Shelf (LCIS; see Fig. 1 for a map of the study area). The LCIS experienced an exceptionally long melt season in 2019/2020[6] and a second peak in surface melting across the AP in February 2022[3], both

[1]Scripps Institution of Oceanography, University of California San Diego, La Jolla, CA, USA. [2]NorthWest Research Associates, Seattle, WA, USA. [3]CIIMAR | Interdisciplinary Centre of Marine and Environmental Research, University of Porto, Porto, Portugal. [4]British Antarctic Survey, Cambridge, UK. [5]Byrd Polar and Climate Research Center, The Ohio State University, Columbus, OH, USA. [6]Atmospheric Sciences Program, Department of Geography, The Ohio State University, Columbus, OH, USA. [7]Antarctic Meteorological Research and Data Center, Space Science and Engineering Center, University of Wisconsin-Madison, Madison, WI, USA. [8]Department of Physical Sciences, School of Science, Technology, Engineering, and Mathematics, Madison Area Technical College, Madison, WI, USA. [9]Institute for Atmospheric and Climate Science, ETH Zurich, Zurich, Switzerland. [10]National Centre for Climate Research (NCKF), Danish Meteorological Institute, Copenhagen, Denmark. [11]Institute of Atmospheric Physics (IAP), Chinese Academy of Sciences (CAS), Beijing, China. ✉e-mail: x4zou@ucsd.edu

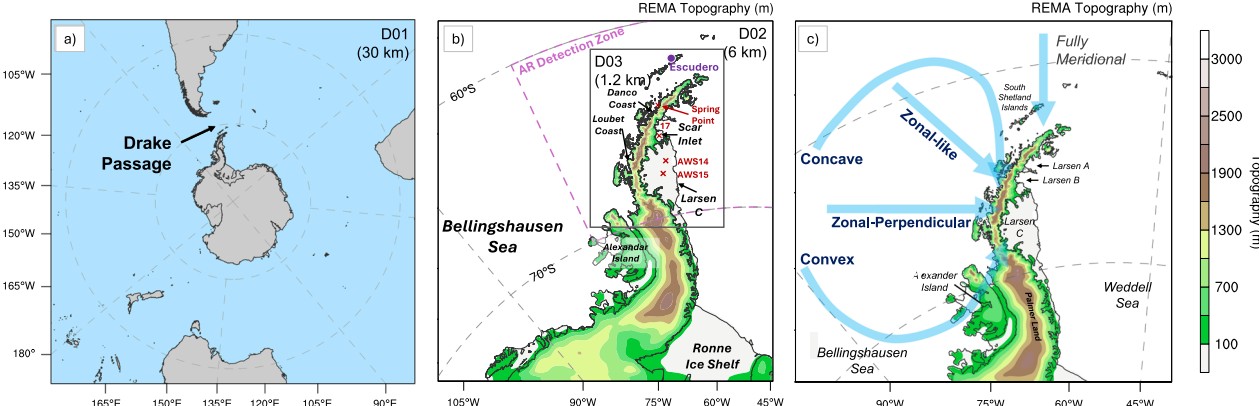

**Fig. 1 | PWRF model domains and an overview of four atmospheric river (AR) shapes. a** D01 with a horizontal resolution of 30 km; **b** D02 and D03 with resolutions of 6 km and 1.2 km, respectively; **c** Classification of all events shown in Supplementary Fig. S1 into distinct AR shapes, which are zonal-perpendicular, zonal-like, convex, and concave. **b** The purple dashed line marks the AR detection zone;

Spring Point, AWS14, AWS15 and AWS17 are shown as red crosses, and the Escudero station as a purple dot. Topography in **b**, **c** is from the 6-km PWRF D02 domain based on the Reference Elevation Model of Antarctica (REMA). Larsen C denotes the Larsen C Ice Shelf.

associated with record-high surface temperatures. It is also one of the most vulnerable Antarctic ice shelves to potential surface-melt-induced collapse by 2100, as the projected southward shift of AP temperatures beyond the melt threshold[7] could expose the LCIS to conditions similar to those preceding the Larsen A (1995) and Larsen B (2002) collapses. Any collapse of the LCIS could trigger increased discharge of grounded ice due to the loss of its substantial buttressing capacity, which would result in sea-level rise[8].

On the synoptic scale, atmospheric rivers (ARs) can transport substantial amounts of heat and moisture from lower latitudes to the AP. They often form within Rossby waves[3,9,10], which can be excited by anomalous convection over the central tropical Pacific[10,11]. For example, upstream Rossby wave packets along the polar jet in the Southern Ocean can induce low-pressure anomalies over the Amundsen and Bellingshausen Seas and high-pressure anomalies east of the Drake Passage, promoting AR landfall over the AP[12]. Although ARs are rare events in Antarctica, they are the primary contributors to intense water vapor transport in this region and play an important role in its surface mass balance[9]. For instance, ~60% of the calving events over the AP between 2000 and 2020 were triggered by AR-induced surface melting[9]. Moreover, surface warming over the LCIS can also be significantly influenced by local orographic forcing, and especially from the föhn effect[13]. For example, föhn-induced warming between 1979 and 2018 contributed up to 18% of the annual surface melting over the western region of the LCIS, where it meets the base of the AP[13,14]. The föhn effect can enhance temperatures and surface melting over the LCIS through several physical processes, including isentropic drawdown, latent heat release from windward-side condensation, mechanical mixing, and radiative heating[15]. Also, the western region of the LCIS frequently experiences low-level föhn jets, which are narrow gap flows associated with föhn conditions[14,16]. Föhn jets originating from the eastern AP are typically warmer than the ambient air on the leeside, and have the potential to enhance surface warming[14,17].

Over the AP, the co-occurrence of ARs and föhn events can intensify surface melting and further compromise ice-shelf stability, as demonstrated in recent studies that examined several combined cases[3,18–21]. However, not all AR events lead to widespread föhn warming or extensive surface melting. This is especially the case for the southern LCIS[19], which has been less impacted by recent extreme heat waves over the northern AP that are related to the compounding effects of ARs and föhn[22,23]. For example, the magnitude of föhn-related temperature increases is primarily influenced by the upstream

atmospheric conditions that are responsible for the associated "flow over" regime[13,15,17,23]. Yet how these conditions are influenced by synoptic circulation patterns such as ARs is poorly understood[19,23].

A key unanswered question is, therefore, how varying characteristics of ARs influence föhn mechanisms over the LCIS and, consequently, its surface warming patterns. For instance, given that föhn-induced warming over the LCIS is a consequence of upstream flow encountering the steep orographic barrier of the AP[17,19,23,24], the characteristics of ARs—such as their shape (e.g., straight or curved), vertical profile (e.g., height of AR core), direction, landfall location, frequency, strength, and duration—are major considerations that could affect this. Furthermore, the abundant moisture transport associated with ARs influences cloud radiative effects, a key driver of surface melting[24]. However, while prior AR studies have primarily examined their frequency over the AP[25], substantial uncertainty remains regarding how AR shape, vertical structure, and landfall location influence föhn-induced surface warming over the LCIS. This study addresses this research gap using high-resolution Polar WRF (PWRF) simulations (up to 1.2 km) to examine 37 austral summer AR events over the AP from 2001–2022 (Fig. 1a, b), providing a representative sample of AR characteristics. This resolution adequately resolves terrain-modified flow over the complex AP topography, which is essential for capturing the occurrence and intensity of föhn-induced surface warming[17] and is poorly represented in coarse-resolution global reanalysis datasets (~25–50 km)[3,21,26].

## Results

### Identification of distinct AR shapes associated with föhn-induced warming over the LCIS

Using the gridded Enhanced AR Scale dataset that we developed for this study[27,28], which is based on integrated vapor transport (IVT) fields calculated using ERA5 output (see Methods and Supplementary Materials), we identified 38 AR events during austral summers from 2001 to 2022 that occurred over the AP region[29] (Supplementary Table S1). Each of the 38 ARs had a ranking that varied from 1 to 3 on the Enhanced AR Scale, which for Antarctica indicates moderate to strong AR events (Supplementary Table S1)[28,30]. Of these events, 37 made landfall on the upwind/western side of the AP, while one AR event made landfall on the LCIS and was subsequently removed from the study. Thus, our following study focuses on the 37 events that made landfall upstream (Supplementary Fig. S1 and Table S1). These included four events with sequences of back-to-back landfalling ARs, referred to as AR-family events (Supplementary Fig. S1 and Table S1).

Consistent with previous studies, 8 out of these 37 events occurred after 2020, reflecting the recent increase in reported extreme weather events over the AP region[3,19,20,22]. All 37 AR events are simulated using PWRF with a horizontal resolution of up to 1.2 km (comprising a total of 3120 h of PWRF simulations).

Initial examination of 6-hourly IVT and mean sea level pressure (MSLP) snapshots based on PWRF outputs from all 37 AR events[29] reveals five distinct AR shapes over the AP region (Fig. 1c and Supplementary Fig. S1, Tables S1 and S2): zonal-perpendicular (promoting perpendicular airflow toward the mountain range upstream of the LCIS), zonal-like, concave (cyclonically curved), convex (anticyclonically curved), and fully meridional (nearly straight north-south airflow). Note that most of the AR events exhibited more than one shape due to changes in regional circulation. To focus specifically on the co-occurrence of ARs and föhn events over the LCIS associated with the different AR shapes, we subsequently select a subset of AR events that resulted in the LCIS experiencing föhn-induced surface warming and classify them according to the Enhanced AR Scale[28] (see Methods and Supplementary Materials for details; Supplementary Table S1). Among the 37 AR events, 23 events exhibited identifiable föhn-induced warming periods over the LCIS (comprising 1165 h of PWRF simulations; hereafter referred to as "AR-föhn events"), including three events classified as AR1, fourteen as AR2, and six as AR3 (the latter including three prolonged AR-family events). Not surprisingly, the AR-föhn events associated with föhn-induced warming over the LCIS tended to make landfall over the central or southern AP (Supplementary Fig. S1). The average warming over the LCIS associated with the 23 AR-föhn events, defined as the increase in hourly maximum 2 m temperature ($T2_{max}$) after removal of the diurnal cycle, varies from ~4.1 °C for AR1 events to 5.6 °C for AR2 events and 7.1 °C for AR3 events (Supplementary Table S1).

Next, cluster analysis[31] of the 500 hPa geopotential height and IVT fields from the simulations of the 23 AR-föhn events (1165 h) is used to investigate the distinct AR shapes associated with these, which confirms the importance of zonal-perpendicular, zonal-like, convex, and concave AR shapes (Fig. 2). The fully meridional AR shape was discarded from our analysis due to its infrequent occurrence and relatively smaller impact on the LCIS[29]. For this AR shape, the regional flow is mostly parallel to the AP, so the flow component perpendicular to the AP[23], and hence the development of föhn, is relatively limited over the LCIS. Consequently, the results presented hereafter focus on four AR shapes: zonal-perpendicular, zonal-like, convex, and concave (Fig. 2).

The cluster analysis shows that AR-föhn events over the LCIS are most frequently driven by zonal-perpendicular (45%) and zonal-like AR shapes (21%), associated with a coupled low-high pressure system (Fig. 2a, b, e, f). These are followed by convex AR shapes (19%) associated with an upwind blocking high over the AP (Fig. 2c, g), and concave AR shapes (15%) that are more likely influenced by a low-pressure system near Alexander Island (Fig. 2d, h). The linkage between AR shape and synoptic circulation is stronger for zonal-perpendicular and convex AR shapes, and weaker for zonal-like and concave AR shapes. Moreover, the IVT cluster analysis indicates that zonal-perpendicular and zonal-like AR shapes that trigger the föhn-induced warming over the LCIS often make landfall over the central AP (Fig. 2e, f), convex AR shapes favor landfall over the southern sector (Fig. 2g), whereas concave AR shapes tend to make landfall further north (Fig. 2h). Furthermore, concave AR shapes exhibit the highest IVT magnitudes (>650 kg m⁻¹ s⁻¹; Fig. 2h), followed by zonal-like AR shapes (>500 kg m⁻¹ s⁻¹; Fig. 2f) and convex AR shapes (>400 kg m⁻¹ s⁻¹; Fig. 2g). Zonal-perpendicular AR shapes exhibit the lowest IVT among all AR shapes (~200 kg m⁻¹ s⁻¹; Fig. 2e). We also note that while the cluster analysis may not fully distinguish all AR shapes, it clearly separates convex and zonal-like AR shapes, likely due to their distinct circulation characteristics.

## Föhn-induced surface warming patterns over the LCIS associated with each AR shape

We then analyze hourly rates of change in 2 m temperature (T2) from the simulations based on the IVT clusters to investigate the surface warming characteristics over the LCIS associated with each AR shape (Fig. 3a–d). Note that the diurnal cycle is removed to better isolate the combined impacts of ARs and föhn-induced warming. The zonal-perpendicular, zonal-like, and convex AR shapes all produce distinct spatial patterns of föhn-induced warming across the LCIS; zonal-like ARs yield the strongest warming in the north, followed by convex ARs with stronger warming in the south. In particular, the zonal-perpendicular AR shape, despite being theoretically expected to produce the strongest föhn warming over the LCIS due to near-perpendicular flow across the mountain barrier, results in only moderate temperature increases (see case study below). The zonal-like AR shape was associated with the highest positive temperatures over the northern LCIS along the edge of the AP mountain chain. This is partly due to the curvature of the AP towards the east at its northern tip (Fig. 1), which means that zonal-like AR shapes making landfall over this region have relatively strong flow impinging nearly perpendicular to the mountain barrier (see their ref.[19] Fig. S4), implying more flow-over regimes and subsequently stronger föhn warming. Despite having lower IVT intensity than concave and zonal-like ARs, the convex AR shape still favors strong warming over the southern sector (Fig. 2), highlighting the unique impact of this AR shape (see case study below). The concave AR shape, however, produces the weakest föhn warming, confined to the northern LCIS near the ice-shelf edge. This is likely owing to its landfall location and the curved geometry of the AP, which enhances mountain blocking and thereby limits the extent of föhn warming.

To further diagnose the surface energy balance (SEB) associated with these warming patterns, we analyzed hourly averaged SEB components based on IVT clusters. While this approach does not directly quantify the rate of temperature change during ARs and föhn-induced warming, it helps elucidate the dominant warming patterns over the LCIS. Consistent with previous studies showing that isentropic drawdown, mechanical mixing, and radiative heating are key processes driving föhn-induced warming over the AP[17], sensible heat flux (SHF) together with downward SW and downward longwave (LW) radiation dominate the SEB over the LCIS (Fig. 3)[13,19,32], i.e., more important than other components like latent heat flux (Supplementary Fig. S2a–d). Downward SW radiation exhibits a strong diurnal cycle, with peak values >800 W m⁻² and the hourly mean contribution of ~220 W m⁻² (Fig. 3e–h). Net SW radiation shows an overall positive contribution (Supplementary Fig. S2e–h). In contrast, downward LW radiation provides a more sustained contribution to surface warming over the LCIS (~300 W m⁻²; Fig. 3i–l); however, net LW radiation remains negative due to enhanced LW emission from the warming surface (Supplementary Fig. S2i–l).

For the zonal-perpendicular AR shape (i.e., moderate warming over the entire LCIS), downward SW radiation contributes ~230 W m⁻² over the northern LCIS, particularly near the ice-shelf edge adjacent to the ocean, while SHF provides a more moderate contribution that is largely confined to the mountain foothills (Fig. 3e, m). Downward LW radiation contributes >270 W m⁻² to the southern LCIS near the mountain foothills (Fig. 3i). For the zonal-like AR shape (i.e., strong warming over the entire LCIS, particularly over the north), downward SW contributions are similar to those in the zonal-perpendicular AR shape, while downward LW radiation and SHF are stronger, likely responsible for the warming pattern (Fig. 3f, j, n). The substantial SHF contribution for the zonal-like AR shape near the mountain foothills over the northwestern LCIS (exceeding 100 W m⁻²; Fig. 3n) is likely driven by enhanced isentropic drawdown and mechanical mixing in response to stronger föhn-induced downslope winds[13]. For the convex AR shape (i.e., strong warming over the entire LCIS, particularly over

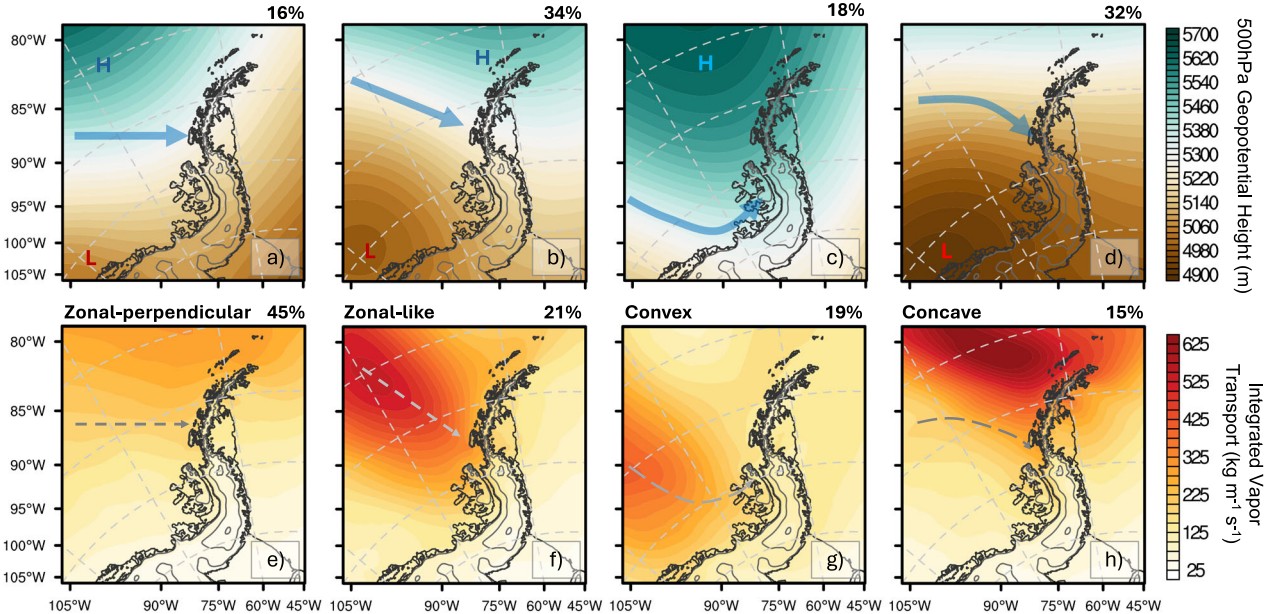

**Fig. 2 | K-means clustering results for synoptic circulation and atmospheric river (AR) shape. a–d** 500 hPa geopotential height (m) and **e–h** Integrated vapor transport (IVT; kg m⁻¹ s⁻¹) based on output from the PWRF simulations (D02: 6 km) of the 23 AR-föhn events that produced identifiable föhn-induced warming over the LCIS (1165 h of simulations). The four AR shapes identified by the cluster analysis are zonal-perpendicular (**a**, **e**), zonal-like (**b**, **f**), concave (**c**, **g**) and convex (**d**, **h**), which are illustrated by the blue arrows and white dashed lines. The red "L" marks low-pressure systems, and the blue "H" marks high-pressure systems. The percentage differences arise because the 500 hPa geopotential height and IVT cluster analyses were conducted independently.

the south), downward SW radiation exhibits relatively high values, particularly over the northern LCIS (exceeding 250 W m⁻²; Fig. 3g). Downward LW radiation is moderate (-240 W m⁻²; Fig. 3k). In addition, the strongest SHF occurs near the mountain base over the central and southern LCIS (approaching 200 W m⁻²; Fig. 3o), forming stripe-like patterns that may be associated with gap flows. Finally, for the concave AR shape (i.e., weak warming over the northern LCIS), moderate downward SW values occur over the northern LCIS, although values are higher over its northern sector (up to 270 W m⁻²) compared to its southern sector (less than 200 W m⁻²; Fig. 3h). Downward LW radiation is the strongest among all AR shapes (up to 300 W m⁻²; Fig. 3l). Moreover, SHF values associated with this AR shape are generally low across the LCIS (-20 W m⁻²) but increase substantially over its northern edge (-80 W m⁻²; Fig. 3p). Although SEB components exhibit relatively high values, indicating a warmer background state, AR-induced föhn warming has limited contribution under the concave AR shape.

**How does the AR shape influence Föhn conditions and warming over the LCIS?**

To better understand how the physical processes associated with föhn warming over the LCIS are influenced by the AR shape, we now examine two representative AR-föhn events featuring zonal-perpendicular and convex shapes for detailed analysis (Figs. 4–7). This approach enables a clearer differentiation of the key föhn processes associated with a particular AR shape, which is slightly limited using the cluster analysis due to the different shapes not always being well distinguished from each other, especially for the zonal-perpendicular shape. Additionally, these two AR shapes are chosen because of their widespread AR-föhn impacts over the LCIS, which is especially apparent for the convex shape (Fig. 3). Selection of these two events as representative AR-föhn events[29] for these categories was based on examination of 6-hourly IVT and hourly T2 fields (Supplementary Table S1), which show that both are three-day AR2 events, associated with pronounced föhn-driven warming over the LCIS during these periods. The zonal-perpendicular AR-föhn event occurred

from 1–3 Dec 2008, and the convex AR-föhn event from 22–24 Feb 2013. Notably, the 2013 convex event exhibited the strongest föhn warming over the LCIS among all convex-dominated AR events (T2$_{max}$ increased by 8.4 °C; Table S1) and ranked third among all events considered. The 2008 zonal-perpendicular event represents a strong föhn warming within its shape category (T2$_{max}$ increased by 4.4 °C; Table S1). Hereafter, these cases are referred to as the "zonal-perpendicular" and "convex" events, respectively.

The zonal-perpendicular AR event made landfall over the central AP with IVT < 500 kg m⁻¹ s⁻¹ (Fig. 4a) and its warm core between 200 and 700 m above sea level showed temperatures of up to 2 °C (Fig. 5a). It was driven by a north-south orientated low-high pressure couplet situated to the west of the AP, which produced strong zonal flow with 10 m wind speeds (W10) exceeding 30 m s⁻¹ upwind of the mountain barrier (Fig. 4a, c and Supplementary Fig. S3a–d). The strong upstream low-level winds associated with the AR (especially its core) decelerated as they approached the AP (Fig. 5a) and were redirected around the orography (Figs. 4c, 5a, and Supplementary Fig. S3a–d), i.e., upstream blocking. Above this layer, the upstream flow was deflected vertically over the AP, resulting in warm and strong downslope winds over the leeside (Figs. 5a, 6a, and Supplementary Fig. S3a–d). Consistent with this, the trajectory analysis shows that nearly all upstream air parcels originating within the elevated AR core reached the leeside (Figs. 4b and 6a, c). This led to widespread warming across the entire LCIS, with T2$_{max}$ of 10.6 °C (Fig. 4c and Supplementary Fig. S3a–d). Further warming from pronounced föhn jets also occurred, which took the form of distinct narrow bands over the central LCIS (Fig. 4c and Supplementary Fig. S3a–d). Upwind precipitation over the AP was relatively strong, likely due to the substantial orographic lift, with 12 h accumulations up to 63.7 mm (Figs. 4d and 5a).

Surface warming over the LCIS during this event became more pronounced after 06Z on Dec 2 (Fig. 7e), which coincided with an increase in maximum SHF (Fig. 7a) and surface downward LW radiation (Fig. 7d). The increase in downward LW over the LCIS is likely due to the broad-scale advection of the AR-associated moisture or cloudiness

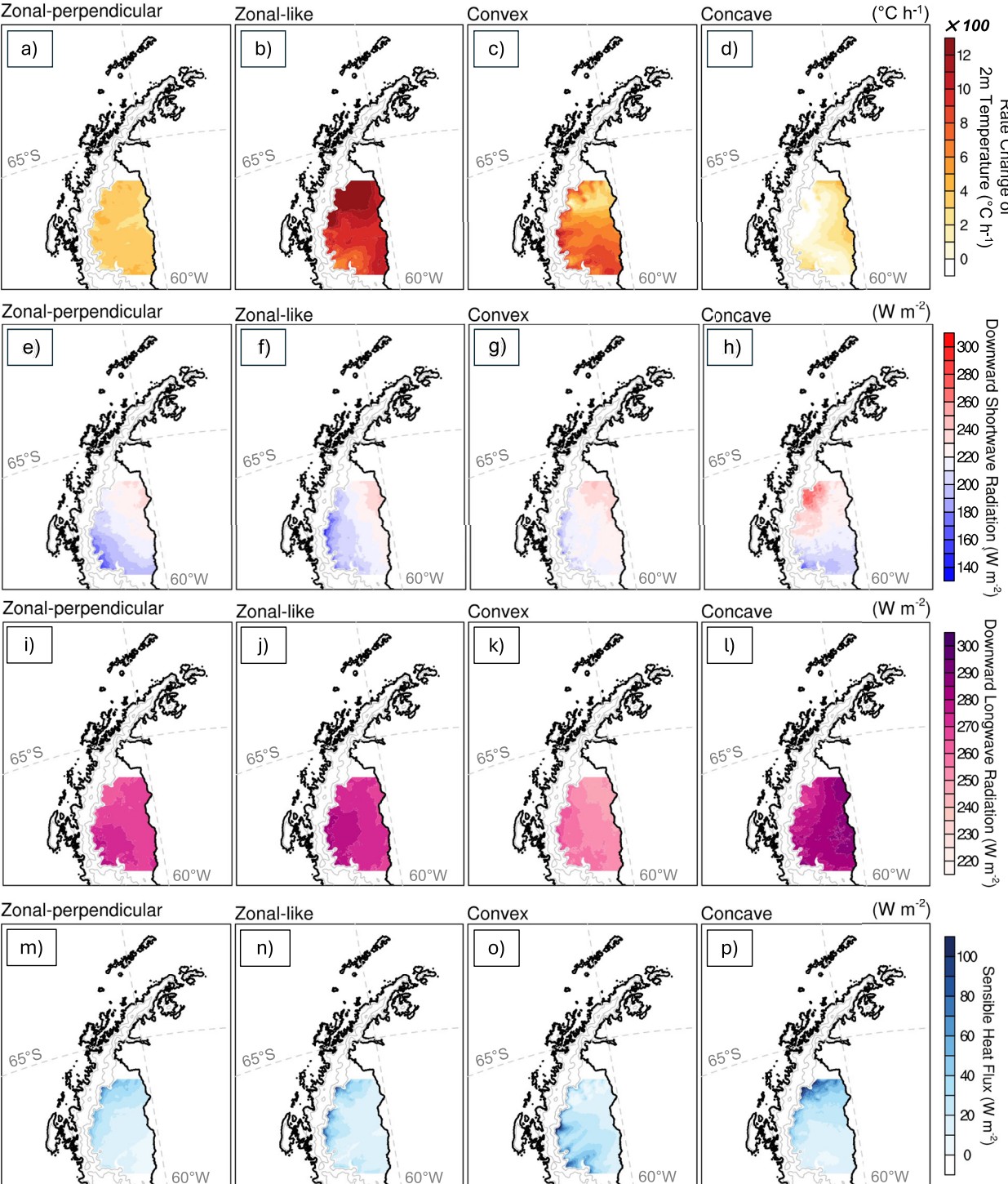

**Fig. 3 | Average value of surface variables in each atmospheric river (AR) shape.** Based on the k-means clustering shown in Fig. 2, composites of (**a**–**d**) the hourly mean rate of change of 2 m temperature ($\frac{dT2}{dt}$; × 100 °C h$^{-1}$), **b**–**h** hourly average surface downward shortwave radiation (W m$^{-2}$), **i**–**l** hourly average surface downward longwave flux (W m$^{-2}$) and (**m**–**p**) hourly average sensible heat flux (W m$^{-2}$) over the Larsen C Ice Shelf for zonal-perpendicular (**a**, **e**, **i**, **m**), zonal-like (**b**, **f**, **j**, **n**), concave (**c**, **g**, **k**, **o**) and convex (**d**, **h**, **l**, **p**) AR shapes. The diurnal cycle has been removed from the 2 m temperature, and values are multiplied by a factor of 100 for visualization.

across the AP (Figs. 4a, 6c and 7b), although this was much reduced near the base of the AP, due to föhn-induced cloud clearance (Supplementary Fig. S4a, b). The highest values of SHF over the LCIS are at the base of the AP (Supplementary Fig. S4c), which is consistent with the föhn-induced downslope winds enhancing turbulent mixing over this region (Fig. 5a). Also noteworthy is evidence that the föhn jets also

introduced enhanced SHF over the LCIS (Supplementary Figs. S3a–d and S4c). These jet areas were associated with a relative reduction in downward SW and an increase in downward LW compared to the rest of the LCIS, which is likely due to the channeling of enhanced AR-associated moisture and cloudiness from the upwind side of the AP through the mountain gaps to the leeside (Supplementary Fig. S4a, b).

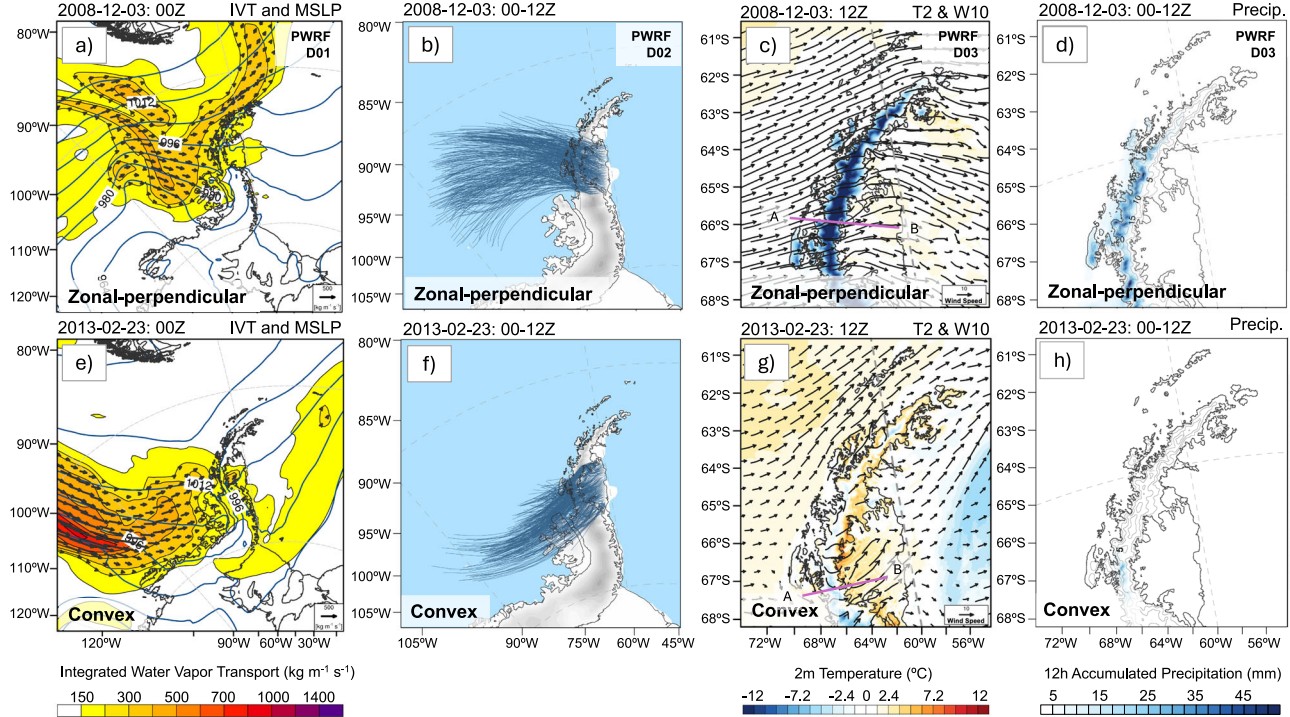

**Fig. 4 | Overview of two representative atmospheric rivers (AR) events.**
**a**–**d** zonal-perpendicular and **e**–**h** convex events, occurring in Dec 2008 and Feb 2013, respectively, and based on PWRF simulations. **a**, **e** Integrated vapor transport (IVT; shading; kg m$^{-1}$ s$^{-1}$) and Mean Sea Level Pressure (MSLP; contours; hPa) at 00Z on 3 Dec and 23 Feb, respectively. **b**, **f** 12-h back trajectories ending over the LCIS from 00 to 12Z on 3 Dec and 23 Feb, respectively. **c**, **g** 2 m temperature (T2; °C) and 10 m wind fields (W10; m s$^{-1}$) at 12Z on 3 Dec and 23 Feb, respectively. **d**, **h** 12 h accumulated precipitation (Precip; mm) from 00 to 12Z on 3 Dec and 23 Feb, respectively. The length of the reference arrow in (**a**, **e**) represents the IVT value of 500 kg m$^{-1}$ s$^{-1}$. The length of the reference arrow in (**c**, **g**) represents a wind speed of 10 m s$^{-1}$.

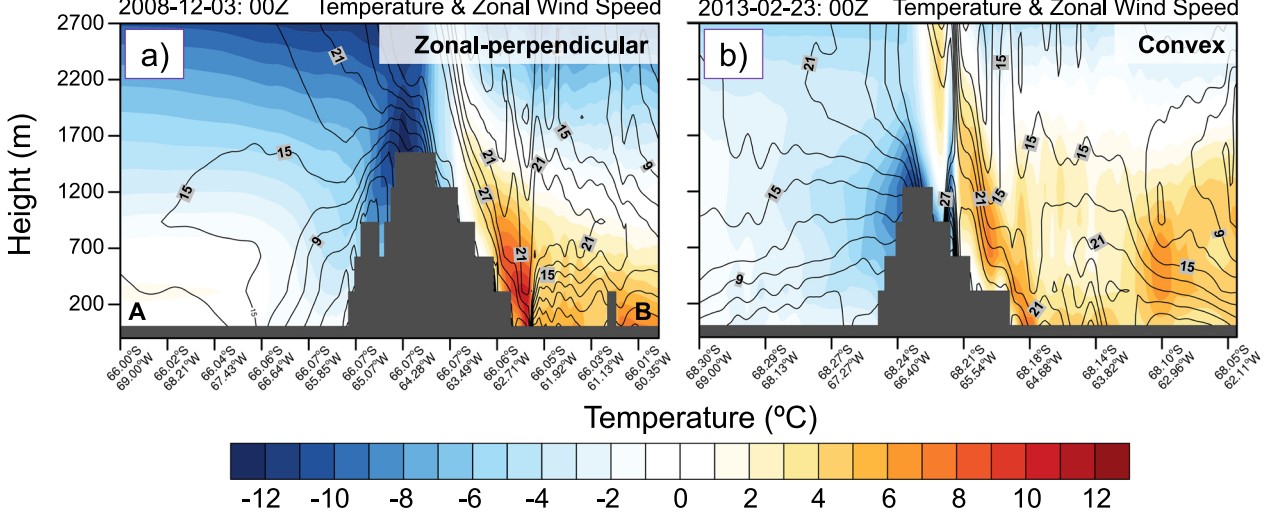

**Fig. 5 | Cross-section of temperature and wind speed for two representative atmospheric rivers (AR) events.** Vertical profiles of **a** zonal-perpendicular and **b** convex AR event, occurring in Dec 2008 and Feb 2013, respectively, and based on PWRF simulations. **a**, **b** temperature (shading; °C) and zonal wind speed (contours; m s$^{-1}$) at 00Z on 3 Dec and 23 Feb, respectively. The location of the cross-section is shown by the purple lines in Fig. 4c, g.

In comparison, the convex AR event made landfall over the southern AP (Fig. 4e) and was driven by a blocking high-pressure system over the Bellingshausen Sea[29]. It also featured higher IVT values (>700 kg m$^{-1}$ s$^{-1}$; Fig. 4e) than the zonal-perpendicular event, and a warm core between 100 and 1000 m above sea level showing temperatures of up to 1 °C (Fig. 5b). This event was associated with markedly less upstream flow being vertically deflected over the AP (Figs. 5b and 6b) compared to the zonal-perpendicular event, and consequently substantial amounts of low-level flow being redirected around the AP (Fig. 4g and Supplementary Fig. S3e–h). However, the comparatively limited vertical deflection was still sufficient to cause warm and strong downslope winds over the leeside (Figs. 5b and 6b), which resulted in a T2$_{max}$ of 14.7 °C over the LCIS (Fig. 4g and Supplementary Fig. S3e–h). Note that the relative lack of vertical deflection over the AP could be related to the elevated warm core of the AR (Fig. 5b), resulting in very stably stratified conditions upstream, as well as the presence of the blocking high-pressure system. Additionally, the weaker orographic lifting of the moist upstream flow resulted in the

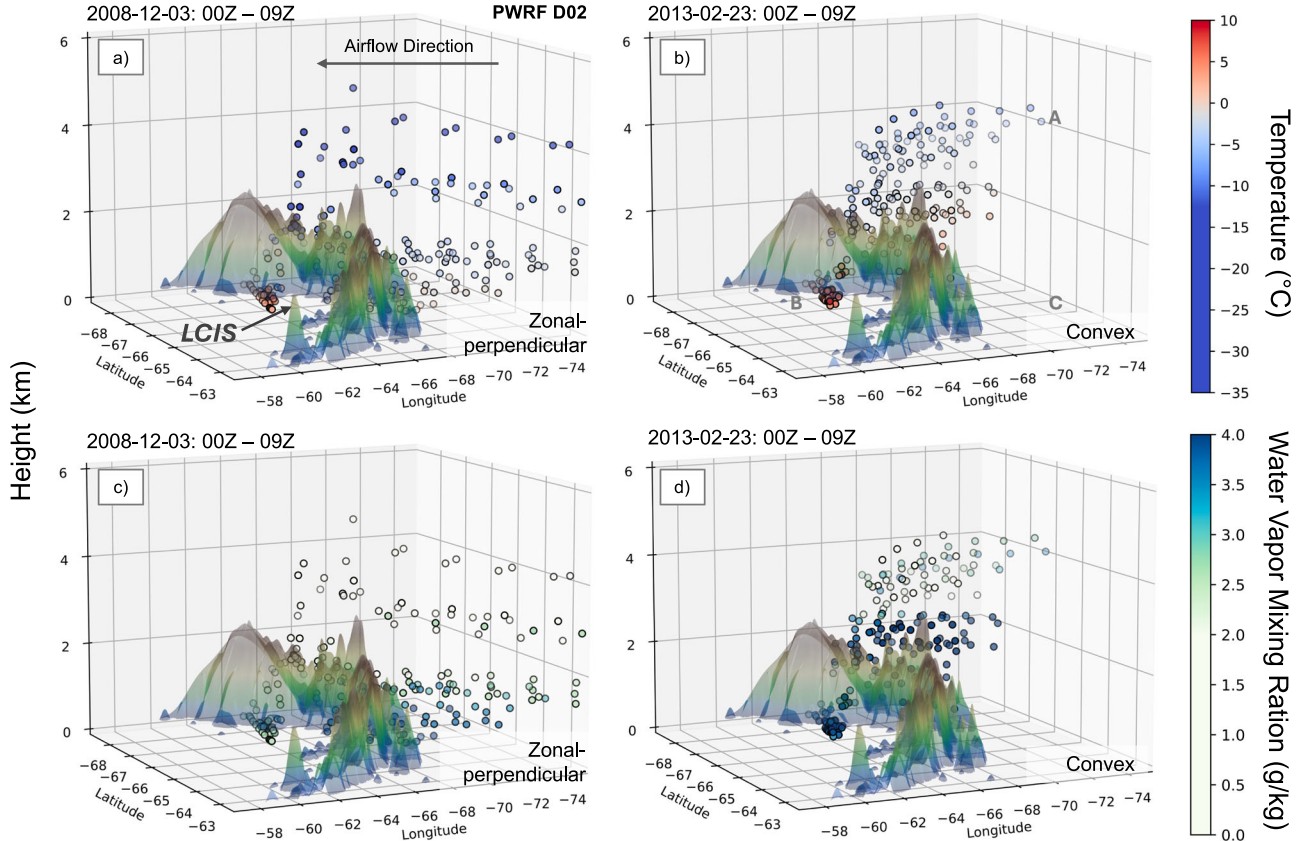

**Fig. 6 | Three-dimensional trajectories of temperature and moisture for two representative AR events.** Nine-hour backward trajectories that initialized 2 m above the Larsen C Ice Shelf (LCIS) surface of (**a**, **c**) zonal-perpendicular and (**b**, **d**) convex events, occurring in Dec 2008 and Feb 2013 respectively, and based on PWRF simulations. **a**, **c** trajectories showing temperature (°C) and water vapor mixing ratio changes (g kg⁻¹) from 00Z to 09Z on 3 Dec. (**b**, **d**) as (**a**, **c**) but from 00Z to 09Z on 3 Feb. **b** Point A, B and C are used for quantification of each föhn mechanism. Nine-hour backward trajectories are initiated from Point B and traced to Point A. Point C lies directly beneath Point A (sharing the same latitude and longitude) but at the same elevation as Point B. LCIS in (**a**) refers to the Larsen C Ice Shelf. All values are archived in the source data.

moderate precipitation over the upstream side of the AP, with 12 h accumulations up to 29.5 mm (Figs. 4h and 5b).

Surface warming over the LCIS during this event exhibited an ~15 °C median temperature increase within 12 h, starting around 06Z on Feb 22 (Fig. 7j). This rapid warming was accompanied by a peak in maximum SHF, enhanced surface downward LW, and reduced surface downward SW (Fig. 7f–i and Supplementary Fig. S4d, e), which suggests that it's due to a combination of föhn effects as well as the broad-scale advection of the AR-associated warmth, moisture, and cloudiness across the AP (Figs. 4e and 6b, d). Because SHF exhibited strong spatial variability and affected only limited areas, its median value remained relatively stable during the event. Over the southern LCIS, where AR-associated moisture exerted the strongest influence, enhanced downward LW could exceed the contribution from downward SW[29]. Compared to the zonal-perpendicular event, the convex event notably produced stronger warming not only near the surface but also throughout the lower troposphere, reaching up to 2 km above the surface and extending farther from the mountain base (Fig. 5b), which is also consistent with the AR itself having a substantial impact on the leeside of the AP. Föhn jets were also present over the southern LCIS during this event with limited influence on the northern sector (Supplementary Fig. S3e–h, and S4f).

Finally, we calculated the total föhn-induced warming over the LCIS within a 9 h window for both the zonal-perpendicular and convex events, along with the relative contributions from individual föhn mechanisms to further quantify the underlying dominant processes (Table 1). Here, föhn-induced warming is associated with a median

temperature increase of 5.1 °C over the LCIS for the zonal-perpendicular event, and a slightly stronger warming of 5.6 °C for the convex event. In both events, isentropic drawdown is the dominant mechanism, contributing a median value over the LCIS of 15.3 °C for the zonal-perpendicular event and 22.8 °C for the convex event, i.e., consistent with the convex event ultimately producing stronger warming than the zonal-perpendicular event. Latent heat release from windward-side condensation plays a secondary role in both events, contributing a median value of less than 1 °C warming over the LCIS, which is perhaps consistent with much of the AR-associated moisture reaching the LCIS through mountain gaps. However, in both cases, there was also a substantial contribution over the LCIS from the combined radiative and sensible heat flux term, which is negative and thus offset the isentropic drawdown contribution, with median values of −8.5 °C for the zonal-perpendicular event and −14.9 °C for the convex event. The large negative values for both events are consistent with the broad-scale advection of the AR-associated warmth, moisture, and cloudiness across the AP, resulting in a reduction in downward SW, despite the positive contributions from the SHF.

In addition, this study also briefly examined representative events for the concave and zonal-like AR shapes, which occurred in Feb 2003 and Feb 2022, respectively[3,19]. Both events made landfall over the northern AP, with IVT values around 1000 kg m⁻¹ s⁻¹. The associated surface warming was therefore mainly confined to the northern AP, especially around Scar Inlet. Both events were also associated with strong upwind precipitation, with 12 h accumulations of 254.6 mm for

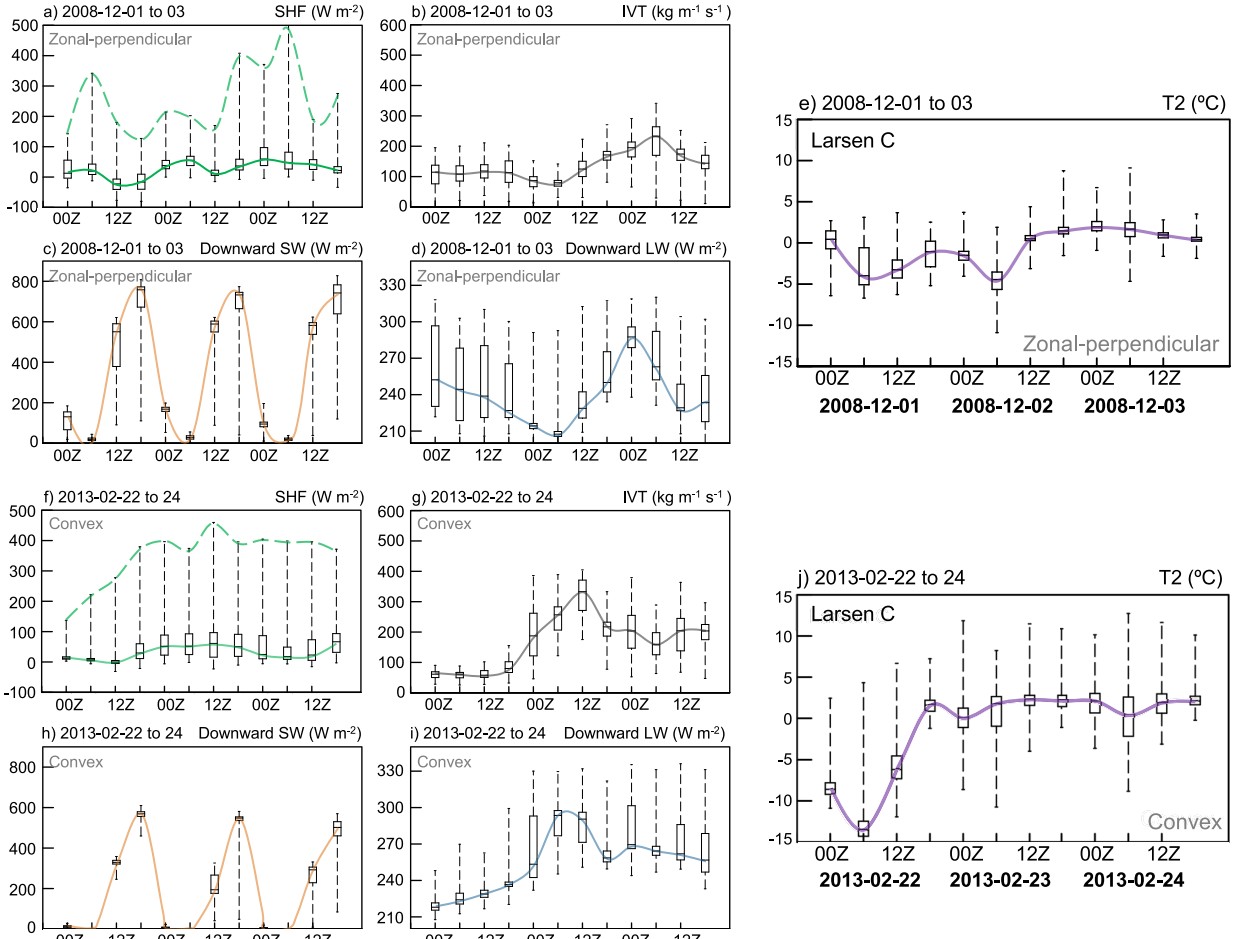

**Fig. 7 | Surface energy balance changes during two representative AR events.** Box plots of six-hourly changes over the Larsen C Ice Shelf of (**a–e**) zonal-perpendicular and (**f–j**) convex AR events, occurring in Dec 2008 and Feb 2013, respectively, and based on PWRF simulations. **a, f** Sensible Heat Flux (SHF; W m⁻²), **b, g** Integrated vapor transport (IVT; kg m⁻¹ s⁻¹), **c, h** surface downward shortwave radiation (Downward SW; W m⁻²), **d, i** surface downward longwave radiation (Downward LW; W m⁻²), and (**e, j**) 2 m temperature (T2; °C) from 00Z 1 to 23Z 3 Dec and from 00Z 22 to 23Z 24 Feb, respectively. The center line within each box denotes the median (green: SHF; gray: IVT; orange: downward SW; blue: downward LW; purple: T2), while the box bounds represent the upper and lower quartiles (75th and 25th percentiles). The green dashed line in (**a, f**) connects the maximum values across timesteps. All values are archived in the source data.

**Table 1 | Quantification of each föhn mechanism for two representative AR-föhn events (median temperature increase across all trajectories)**

| Mechanisms (median; °C) | Total föhn warming | Isentropic Drawdown | Latent Heat Release | Combined radiative and sensible heat fluxes |
|---|---|---|---|---|
| 2008 zonal-perpendicular event | 5.1 °C | 15.3 °C | 0.5 °C | −8.5 °C |
| 2013 convex event | 5.6 °C | 22.8 °C | 0.6 °C | −14.9 °C |

the concave event and 183.1 mm for the zonal-like event, i.e., consistent with the large IVT values. See Supplementary Fig. S5 for more details.

## Discussion

This study highlights the importance of the shape, vertical profile, direction, and landfall location of ARs for influencing föhn-induced warming over the LCIS by simulating a subset of representative, impactful AR events using a high-resolution regional model. We identified four distinct AR shapes that are associated with föhn-induced surface warming over the LCIS (Figs. 2–4). These are zonal-perpendicular and zonal-like AR shapes (associated with a coupled low-high pressure system), convex shapes (driven by a blocking high), and finally concave shapes (often driven by a low-pressure system). In general, zonal-like and convex AR shapes produce the strongest föhn warming over the

LCIS, followed by zonal-perpendicular ARs. Our results further show that isentropic drawdown, mechanical mixing, and radiative heating were key components of föhn-induced warming over the LCIS (Table 1 and Figs. 4–6). Elevated SHF is apparent around the base of the AP (Fig. 3), likely driven by enhanced isentropic drawdown and mechanical mixing in response to stronger föhn-induced downslope winds, as well as föhn jets (Figs. 3 and 4). Increased surface downward SW radiation along the LCIS margin is consistent with föhn-induced cloud clearing, while enhanced AR-related moisture and downward LW radiation near mountain gaps (Fig. 3) can locally exceed the SW contribution.

We introduced the AR shape as an effective indicator for AR-föhn analysis over the LCIS. Previous AR research has largely emphasized intensity and duration[33], with fewer studies addressing features such as AR shape[34,35]. AR shape is tightly coupled to synoptic circulation and

encapsulates multiple AR characteristics, thereby impacting the magnitude of föhn warming by governing the ability of the incoming airflow to cross mountain barriers. In this study, we showed that each AR shape resulted in a distinct pattern of warming over the LCIS (Figs. 2 and 3). For zonal-perpendicular and zonal-like AR shapes making landfall over the central LCIS, the zonal-like one is generally characterized by higher moisture content and stronger winds due to its lower-latitude moisture source and the presence of a coupled low-high pressure system (Fig. 2). The convex AR shape tends to make landfall farther south and promotes intense downslope winds over the LCIS under the influence of a blocking high (Figs. 5 and 6). Thus, although the airflow crosses relatively lower topography over the southern AP, a strong föhn effect occurs due to pronounced adiabatic warming (Table 1). This result provides a detailed explanation for why blocking highs can strongly promote surface warming over the AP[36,37]. The concave AR shape, despite exhibiting the strongest IVT magnitude, produces a limited föhn response over the LCIS because its curved flow favors a more northerly landfall location and is largely restricted to the western side of the AP (Supplementary Fig. S5a–d). Recent studies have begun to emphasize the role of AR curvature in affecting surface impacts, a concept closely aligned with the AR shape framework introduced here[38].

Moreover, our results suggest that the flow regime over the AP was also influenced by vertical profiles such as their elevated warm and moist cores, which led to very stably stratified conditions[21]. These conditions would encourage upstream blocking and redirection of the flow around the orography[23,24], rather than a flow-over regime (Figs. 4–6). Additionally, the broad-scale advection of the AR-associated warmth, moisture, and cloudiness across the AP was also noticeable over the eastern side of the AP, apparent by the occurrence of stronger warming not only near the surface but also throughout the lower troposphere (Fig. 5).

This study has several limitations that should be addressed in future research. First, there are uncertainties arising from differences among AR detection algorithms[39,40]. The detection of AR depends strongly on the choice of moisture threshold (fixed versus relative), the AR identifier (IVT, using meridional-only or combined zonal and meridional components), and the datasets employed[40,41]. Among these, the moisture threshold is the dominant source of uncertainty, significantly affecting the interpretation of AR frequency[42]. Although our selection is consistent with an independent global dataset that applies the Elongated Targets (tARget) algorithm[43] (e.g., Supplementary Fig. S6), further research aimed at reducing uncertainties in AR detection would be beneficial. Notably, three of the four AR-family events belong to AR3, warranting further investigation of this category and its compound impacts. Second, our analysis is limited to post-2001 events, excludes weaker ARs, and applies relatively strict föhn-warming criteria confined to the LCIS (e.g., hence why only 23 of the 37 AR events initially analyzed exhibited identifiable föhn-induced warming periods). Although the selected subset is sufficiently representative, the development of more comprehensive datasets of AR-induced föhn events and föhn-only events along the Antarctic coastline would be beneficial. Third, the cluster analysis employed in this study robustly identifies convex and zonal-like AR shapes and their associated synoptic circulation patterns, whereas concave and zonal-perpendicular AR shapes are more difficult to fully distinguish. These latter two are more sensitive to the choice of AR-induced föhn-warming periods, and their linkage to specific synoptic circulation patterns is weaker. For example, several zonal-perpendicular AR events occur under the influence of a strong low-pressure center positioned near Ellsworth Land, instead of a coupled low-high pressure system. More advanced machine-learning techniques, such as self-organizing maps[44], may help better characterize AR shape and curvature. In sum, further research is still needed to fully understand Antarctic AR climatology and associated characteristics.

The complex relationship between ARs and föhn also emphasizes the need for high-resolution coupled model simulations and observations of clouds, the SEB, and precipitation over Antarctica[26]. Incorporating AR characteristics is essential for assessing surface impacts along the entire Antarctic coastline, as combined AR-föhn warming has been reported beyond the AP[45]. This compound impact not only affects ice-shelf stability but may also trigger the disintegration and advection of sea ice in the downstream[20]. Different AR configurations, characterized by distinct wind regimes and thermodynamic conditions[46], together with oceanic forcing[47], may be key to understanding sea-ice coverage and thickness over the AP. Under a warming and more moisture-rich future climate, stronger ARs may penetrate farther inland, facilitating greater moisture intrusion and more intense winds along the coast of the Antarctic continent[48]. A higher probability of stronger ARs could consequently result in more frequent and severe surface warming events[49,50] and sea-ice retreat[46,51]. Therefore, on individual ice shelves along the Antarctic coastline, those regional-scale atmospheric processes that are poorly resolved in global climate models will become increasingly important for ice-shelf stability[52] and sea-ice conditions[53,54], ultimately influencing Antarctic ice loss and global sea-level rise. This further highlights the need to develop high-resolution regional coupled models that explicitly resolve interactions among Earth system components.

## Methods

### Reanalysis data and observations

The 5th major atmospheric reanalysis (ERA5)[55] produced by the European Center for Medium-Range Weather Forecasts, was used to develop the gridded Polar AR dataset used to select AR events[27]. This study also used the ERA5 output, pre-processed by the NSF National Center for Atmospheric Research (NCAR), as input to drive the high-resolution PWRF simulations (see below)[56].

Downward LW and SW were measured at Escudero Station (62.2°S, 58.97°W), which is situated north of the AP on King George Island. Downward LW was measured with a Kipp & Zonen pyrgeometer, and downward SW using a Kipp & Zonen SMP21-V global hemispherical irradiance pyranometer. More details on these measurements and calculation of clear sky downward radiative fluxes are given in previous research[57,58]. Observations from automatic weather stations situated on the LCIS from AWS14 (67.02°S, 61.5°W, northern LCIS), AWS15 (67.57°S, 62.15 °W, southern LCIS), Spring Point (64.29°S, 61.05°W), and AWS17 (65.93°S, 61.85°W) are also used to evaluate PWRF's performance near the surface (see Fig. 1b for site locations).

### PWRF simulations and model evaluation

PWRF, developed and maintained by the Polar Meteorology Group at the Byrd Polar and Climate Research Center, The Ohio State University, is a regional climate model designed for polar regions based on WRF[59,60]. PWRF follows the standard WRF workflow but incorporates multiple updates to its physical parameterizations. This study employed PWRF V4.3.3 to generate hourly output using a downscaling approach with three nested domains: 30 km (domain 1), 6 km (domain 2), and 1.2 km (domain 3) spatial resolutions (see Fig. 1b). All PWRF outputs in this study were initialized at 00Z each day using ERA5 data for surface and lateral boundary conditions, with the first 24 h discarded as spin-up time (only forecasts from 24 h to 47 h were used).

The detailed physical parameterization settings are listed in Table S3. The two-moment Morrison-Milbrandt P3 (P3) scheme was selected for its reliable estimation of liquid water path and LW radiation[61,62]. The Mellor-Yamada-Nakanishi-Niino (MYNN) scheme[63] was used for the atmospheric boundary layer. Both LW and SW radiation were represented by the Rapid Radiative Transfer Model (RRTMG)[64]. The Kain-Fritsch scheme[65] was applied for the parameterization of convection in domain 1 at 30 km spatial resolution, but was switched off for the other higher-resolution domains as

convection is resolved explicitly. The Noah-MP model[66] scheme was used for land surface processes.

In addition, PWRF simulations incorporated surface albedo observations from the Moderate Resolution Imaging Spectroradiometer (MODIS) and topography data from the Reference Elevation Model of Antarctica (REMA)[67] to better capture surface changes and improve the representation of airflow modifications by local topography[68]. The MODIS albedo, produced from 16 days of Terra and Aqua observations by the National Aeronautics and Space Administration (NASA), has been validated in surface melting studies over West Antarctica, demonstrating improved surface temperature estimation[69,70]. Incorporating the MODIS albedo also helped reduce the systematic cold bias over the Tibetan Plateau in WRF[71]. REMA provides a high-resolution surface elevation dataset with up to 8-meter spatial resolution, derived from stereophotogrammetry and satellite imagery[67]. This model setup has been extensively tested in previous Antarctic AR studies and demonstrated robust performance, particularly in simulating SEB and temperature[3,19,57,72–74].

To further assess the accuracy of the PWRF model for this study, output from the simulation of the AR event from 22 to 24 February 2013 was compared to standard meteorological observations at AWS14 and AWS15 on the LCIS. This comparison showed that PWRF relatively accurately captured the temperature increase over the LCIS associated with föhn warming, with an average bias of less than −1.5 °C (Supplementary Fig. S7). However, like other high-resolution models[14], PWRF exhibits a larger bias in near-surface wind, likely due to model instability. Additionally, downward SW, downward LW, and total radiative fluxes from ERA5 and PWRF are compared to observations[75,76] during an AR2 event in January 2022 (Supplementary Fig. S8). Clear-sky simulations are also shown from ERA5 and (for downward LW only) based on simulations using radiosonde profiles. PWRF outperforms ERA5 in both downward SW and LW at Escudero for this event, and consistently demonstrates superior performance across several other AR events during the Year of Polar Prediction in the Southern Hemisphere (YOPP-SH) campaign period[30,57], likely due to its more advanced microphysical scheme and higher spatial resolution. Additional model evaluations of surface meteorological variables, based on available AWS observations during AR events, are summarized in Supplementary Table S4. Future research will help further improve PWRF simulations by exploring alternative reanalysis datasets for initial and boundary conditions, refining model physical parameterizations, and incorporating additional observations.

## Methodology overview

In general, this study was conducted following these steps to select and investigate AR-föhn events: 1) used the gridded Enhanced AR dataset[27] to identify all AR events influencing the AP from 2001 to 2022 during austral summer, and categorized them based on AR shapes[29]; 2) performed high-resolution PWRF simulations for each identified AR cases; 3) analyzed the PWRF outputs to identify model timesteps associated with föhn-driven warming over the LCIS (AR-föhn patterns); 4) conducted an analysis of all AR-föhn patterns; 5) performed detailed case studies of four representative AR-föhn events, for zonal-perpendicular, convex, zonal-like, and concave AR shapes, focusing primarily on the first two. Detailed selection criteria are provided in the Supplementary Materials. This study restricts event selection to years after 2001 because our high-resolution simulations incorporate MODIS albedo observations, which are only available after February 2000.

This study employed k-means clustering analysis and air parcel trajectory analysis. K-means clustering analysis[31] was applied to hourly 500 hPa geopotential height and IVT fields when AR-föhn affected the LCIS. Then, surface variables such as T2 and SEB components were investigated based on IVT-defined clusters. This approach helps characterize föhn-induced surface warming impacts associated with

different AR characteristics, rather than being limited to individual events that may exhibit multiple AR shapes. Notably, the cluster analysis exhibited limitations in completely distinguishing among zonal-like, zonal-perpendicular and concave AR shapes. To address this limitation, we included individual case studies.

For two representative AR-föhn events for zonal-perpendicular and convex AR shapes, which impacted the entire LCIS, we performed trajectory analysis using Read Interpolate Plot (RIP4) with PWRF simulation data as input, allowing us to identify and quantify the contribution of each föhn mechanism under the influence of an intense AR. The RIP4 is a tool developed and maintained by Mark Stoelinga at NCAR and the University of Washington. It tracks air parcel movement and computes meteorological parameters along trajectories. Previously, RIP4 has been used in research yielding reliable results, such as examining the mesoscale topographic influence on tropical cyclone tracks[77] and quantifying leeside föhn warming mechanisms[69].

## Detection and quantification of ARs

To detect ARs, we use the Enhanced AR Scale[28] developed at the Center for Western Weather and Water Extremes (CW3E) at Scripps Institution of Oceanography (Supplementary Fig. S1), which is based on both IVT value (Eq. 1) and the AR duration at a specific location. The regular AR scale[33] ranges from AR1 (weak) to AR5 (exceptional) and is designed for the middle latitudes. In addition to the regular AR scale, the Enhanced AR Scale is extended for the polar regions with lower IVT thresholds of 100, 150, and 200 kg m$^{-1}$ s$^{-1}$, accounting for low-IVT cases in these areas. For Antarctica, a ranking of AR1 to AR3 on the Enhanced AR Scale indicates moderate to strong events (IVT thresholds >250 kg m$^{-1}$ s$^{-1}$). This scale was first tested during the YOPP-SH Winter Targeted Observing Periods and was proven effective in accurately reflecting the strength and duration of ARs[28,30]. See Supplemental Materials for further details (Supplementary Figs. S9 and S10).

In this study, we utilized this Enhanced AR Scale[28] and calculated a new hourly gridded dataset based on IVT from ERA5 reanalysis data. We then computed the regional mean of this gridded dataset within the target area (60°S to 70°S, 45°W to 75°W; indicated by the purple dashed box in Fig. 1 and the red dashed box in Supplementary Fig. S1a). Only AR events with a regional mean AR Scale exceeding AR1 were selected for high-resolution PWRF simulations.

$$IVT = \sqrt{\left(\frac{1}{g}\int_{1000}^{10} q\mathbf{u}\,dp\right)^2 + \left(\frac{1}{g}\int_{1000}^{10} q\mathbf{v}\,dp\right)^2} \qquad (1)$$

where $g$ is the gravity acceleration constant (m s$^{-2}$), $\mathbf{u}$ and $\mathbf{v}$ are zonal and meridional wind components (m s$^{-1}$), $q$ is specific humidity (kg kg$^{-1}$), and $p$ is the pressure (hPa).

## Identification and quantification of Föhn warming

Föhn warming over the LCIS is manually detected based on hourly time series of maximum T2 over the LCIS and 6-hourly snapshots of T2 and W10. Vertical cross-sections of temperature and wind speed were used to further confirm the presence of downslope winds on the leeside. Two key criteria were applied (details see Supplementary Materials):

- The maximum T2 over the LCIS increased by >3 °C for over 6 h after removing the diurnal cycle, with warming detected near the mountain foothills and the leeside significantly warmer than the upwind side. The start of warming is defined as the onset of the maximum T2 increase, and the end is when maximum T2 decreases by >2 °C from its peak (Supplementary Fig. S11).
- The W10 over the LCIS increased during event[21], especially for regions not affected by föhn jets.

No fixed threshold was applied to wind speed changes[21], as multiple cases exhibited gap flows that also led to substantial increases in wind speed. Combined AR-föhn events do not necessarily cause a

decrease in relative humidity (RH) on the leeside, as moisture can reach this region via gap flows or spillover. Therefore, RH changes were not used as a detection metric.

This study also quantified the contribution of each föhn mechanism to surface warming[15] for two representative AR-föhn events, zonal-perpendicular and convex AR shapes. The quantification is based on trajectory analysis. Nine-hour backward trajectories were initiated near the surface over the LCIS (Point B) and traced to the upwind side of the AP (Point A), with associated meteorological variables tracked along the airflow. Similar trajectory-based approaches have been widely used in previous analyses of Antarctic föhn events[15,69]. The corresponding formulations are given in Eqs. (2–8):

$$\text{Föhn Effect (FE)} = T_B - T_C \tag{2}$$

$$\text{Isentropic Drawdown (ID)} = \theta_A - T_c \tag{3}$$

$$\text{Thermodynamic Term (TT)} = (\theta e_A - \theta_A) - (\theta e_B - \theta_B) \tag{4}$$

$$\text{Sensible Heat Flux (SHF)} = \theta e_B - \theta e_A - \int_A^B \Delta\theta \tag{5}$$

$$\text{Radiative Heating (RAH)} = \int_A^B \Delta\theta \tag{6}$$

$$\text{Correction Term (CT)} = T_B - \theta_B \tag{7}$$

$$\text{FE} = \text{ID} + \text{TT} + \text{SHF} + \text{RAH} + \text{CT} \tag{8}$$

where $T$ is temperature, $\theta$ is potential temperature, $\theta_e$ is equivalent potential temperature. Point A, marking the end of the backward trajectory, is located in the undisturbed upstream flow over the ocean, more than ~200 km upstream of the AP—exceeding the typical Rossby radius of deformation (Fig. 6b). Point B, the start of the trajectory, is situated in the region of earliest surface melting over the LCIS, ~2 m above the surface (Fig. 6b). Point C lies directly beneath Point A (sharing the same latitude and longitude) but at the same elevation as Point B, with associated meteorological variables interpolated from the PWRF simulations to this location (Fig. 6b).

## Data availability

The enhanced gridded AR scale dataset generated in this study has been deposited at the Antarctic Meteorological Research and Data Center (AMRDC): https://amrdcdata.ssec.wisc.edu/dataset/gridded-polar-atmospheric-river-ar-scale-dataset. PWRF simulations generated in this study have been deposited at the AMRDC: https://amrdcdata.ssec.wisc.edu/dataset/selected-austral-summer-atmospheric-river-cases-over-antarctic-peninsula-2001-2022. Global Atmospheric Rivers Database V4 is available here: https://doi.org/10.25346/S6/ZSW7UN. ERA5 reanalysis dataset is available at the Copernicus Climate Change Service (C3S) Climate Data Store: https://cds.climate.copernicus.eu/. Reference Elevation Model of Antarctica (REMA) surface topography for PWRF simulations is available at: https://www.envidat.ch/dataset/rema-topography-and-antarcticalc2000-for-wrf. Downward shortwave and longwave radiation collected at King George Island during YOPP-SH have been deposited at AMRDC: https://amrdcdata.ssec.wisc.edu/dataset/shortwave-down-radiative-flux-king-george-island and https://amrdcdata.ssec.wisc.edu/dataset/longwave-down-radiative-flux-king-george-island. Ground observations at AWS14 (67.02°S, 61.5°W) and AWS15 (67.57°S, 62.15 °W) are available at: https://www.projects.science.uu.nl/iceclimate/aws/antarctica.php. Additional 6-hourly AWS observations from AntAWS dataset for Spring Point (64.29°S, 61.05°W) and AWS17 (65.93°S, 61.85°W) are available at: https://amrdcdata.ssec.wisc.edu/dataset/antaws-dataset. SEB data for Fig. 7 are provided in the Source Data file. Source data are provided with this paper.

## Code availability

Data preprocessing and visualization were performed using NCAR Command Language (NCL) and Python. Data visualization was performed using modified NCL and Python plotting libraries. Example codes are available here: https://amrdcdata.ssec.wisc.edu/dataset/selected-austral-summer-atmospheric-river-cases-over-antarctic-peninsula-2001-2022. Source code for RIP4 software used in trajectory analysis is available at https://www2.mmm.ucar.edu/mm5/documents/ripug_V4.html.

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

## Acknowledgements

X.Z. and Z.Z. acknowledge support from NSF-OPP-2229392 and NSF-OPP-2331992. P.R. acknowledges support from NSF-OPP-2127632. This research was supported in part by a grant NSF PHY-2309135 to the Kavli Institute for Theoretical Physics (KITP). I.G. acknowledges support from Portuguese FCT for strategic funding to CIIMAR (UIDB/04423/2020, UIDP/04423/2020), CEEC IND IMPACT (2021.03140.CEECIND/CP1659/CT0005) and project MICROANT (2023.15890.PEX). A.O. is supported by the PolarRES project, which has received funding from the European Union's Horizon 2020 research and innovation program under grant agreement no. 101003590, and the Natural Environment Research Council-funded project PICANTE (Processes, Impacts and Changes of ANTarctic Extreme weather; Grant No. NE/Z503356/1). D.H.B. is supported by NSF-OPP-2205398. D.L. is supported by NSF-OPP-2130203. M.A.L. is supported by NSF-OPP-1951720 and NSF-OPP-2301362. J.D.W. acknowledges support from the European Union's Horizon 2020 project nextGEMS under grant agreement number 101003470. N.H. is funded by the Danish State through the National Center for Climate Research (NCKF) and by the Novo Nordisk Foundation project PRECISE (NNF23OC0081251). The PWRF model is developed and is maintained by the Polar Meteorology Group, Byrd Polar and Climate Research Center (BPCRC), The Ohio State University. PWRF simulations were performed on the San Diego Supercomputing Center's EXPANSE resource through AR Program Phase V 4600015671 (State of California, Department of Water Resources). Contribution number C-1636 of BPCRC. We acknowledge the use of OpenAI's ChatGPT (GPT-5.2, Mar 2026 version; https://chatgpt.com/) to assist with cleaning and commenting example code. All code was subsequently reviewed and validated by the authors prior to deposit at AMRDC.

## Author contributions

X.Z., P.R., and I.G. conceived the project. X.Z. and P.R. conducted the analysis and drafted the initial manuscript. A.O. provided significant input during the analysis and revision process. D.H.B., P.R., I.G., Z.Z., J.W., and D.L. provided valuable suggestions during discussions. Z.Z. contributed to the final proofreading. M.L. and B.K. assisted with data management, while J.L. and P.G. contributed to data visualization. N.H., J.C., and F.M.R. engaged in discussions of the results and offered feedback on the manuscript.

## Competing interests

The authors declare no competing interests.
