## [Transparent Peer Review file · Nature Communications]

Föhn-Induced Melting over Larsen C Modulated by Atmospheric River Shape, Direction and Landfall Location

Corresponding Author: Dr Xun Zou

Version 0:

Reviewer comments:

Reviewer #1

(Remarks to the Author)

This study explores the relationship between atmospheric river (AR) characteristics and foehn-induced surface warming over the Larsen C Ice Shelf (LCIS) in Antarctic Peninsula (AP) using a gridded AR dataset and high-resolution Polar-WRF model simulations for each AR event over 2001-2022. Four distinct AR shapes are identified, and respective physical mechanisms are proposed to explain their impact on foehn warming. The results are novel and significant, which should be of interest to the readership of Nature Communications. The findings have great potential implications for Earth system modeling of Antarctic ice shelf stability and loss. However, there are some technical concerns related to the methodology and analysis. I cannot recommend it for publication in the current form. The following specific comments need to be satisfactorily addressed.

1) When referring to AR shape and landfall location, I would picture the geometric shape of long, narrow moisture corridor upon landfalling over AP, but the schematic illustration in Figure 2 got me confused. Not sure whether the AR pathway represents the moving path of AR. How is the AR pathway for each event identified from the ERA5 data and drawn in Fig. 2a? What do the dates and the length of each line mean?

2) PWRP configuration: How are the model domains aligned with the zonal and meridional bands near AP? There are a couple of related concerns:

a. When nudging u and v winds in the PWRP simulations to 6-hourly ERA5, should the ERA5 grids be remapped and spatially interpolated to the PWRP x and y directions? How?

b. Please clarify how the domain alignments may affect the PWRP simulated AR pathways and shape patterns that are defined relative to the zonal lines. The Fully Zonal AR shape/pathway depicted in Fig. 2b doesn't align with the zonal lines (gray dashed lines).

3) Figure 1: embedding the D01 at the corner of D02, with the shared axes, is very confusing. I suggest moving it out of D02 and marking its own latitude-longitude ranges.

4) The Froude (Fr) number, being used to depict the AP topographic flow blocking, might be able to objectively quantify the impact of AR shape and landfall location on foehn-induced warming. Why was Fr number arbitrarily calculated using the mean U calculated in the small blue box? How was the elevation range 900 m – 1100 m identified?

5) I tend to believe that the AR landfalling location and wind speed (near the height of the topographic elevation range) might be more important to foehn-induced warming than the specific AR patterns (identified by the IVT) and moisture content. Taking the convex AR (2013-02-23), if most of the AR moisture is sourced from distant ocean along an elevated pathway, it's hardly convincing that the AR moisture can have a more significant impact on the LCIS through SHF and downward SW processes than the downward LW radiation. Please comment on this.

6) Reanalysis products can be much biased over polar regions. I wonder if the authors tried a different reanalysis product than ERA5 for driving the PWRP model.

Reviewer #2

(Remarks to the Author)

This study selected cases of Atmospheric Rivers (ARs) that could accompany föhn phenomena in the Larsen C Ice Shelf (LCIS) region from 2001 to 2022 in the Antarctic Peninsula (AP), and investigated the föhn mechanisms based on the trajectory shapes of the ARs. For detailed analysis of these mechanisms, high-resolution downscaling data (1.2 km) were generated using Polar WRF. Using this dataset, the ARs were classified by shape, and the warming patterns in the AP

region were examined. The föhn mechanisms were analyzed in detail for two representative AR cases. Given the recent rapid increase in the frequency of föhn-induced warming events on the Antarctic Peninsula due to accelerated climate change, this study aims to significantly contribute to understanding and addressing Antarctic warming.

Major Concern

This study classified AR shapes using K-means clustering, examined the characteristics of each cluster, and then analyzed the mechanisms of föhn impacts through two detailed AR case studies. The analysis of the föhn mechanisms in these cases is a very important part of the research, as it quantitatively examines surface warming depending on the föhn processes. However, the reason for selecting the fully zonal and convex-shaped ARs, as well as the rationale for choosing the two corresponding cases, is insufficient. It would be necessary to select the target shapes based on at least some quantitative evidence, such as the frequency of each AR or a quantitative comparison of the intensity of warming over LCIS. Similarly, in case selection, providing quantitative selection criteria such as LCIS temperature thresholds would strengthen the justification.

Minor Comments

L40: thereby have -> thereby having a

L116: shapes -> Shapes

L120: Why was the analysis period set from 2001 to 2022? Since ERA5 data cover a longer period, and reanalysis-based studies typically begin from 1979, the reason for choosing 2001 as the starting year should be clarified.

L273: 9 hour -> 9-hour

L308: determines -> determine

In the methods section, a more detailed description of the K-means clustering approach is needed. Was the number of k predetermined?

The term "fully zonal" seems clear in the context of shape classification, but since the trajectories are not strictly parallel to latitude, the use of "fully" may not be linguistically precise.

In k-means cluster analysis, typical information provided in other clustering studies—such as the number of ARs per cluster, mean duration, IVT, and degree of warming—appears to be missing.

Reviewer #3

(Remarks to the Author)

Summary :

In "Föhn-Induced Melting over Larsen C Modulated by Atmospheric River Shape, Direction and Landfall Location," Zou et al. analyze the relationship between ARs and föhn over the Antarctic Peninsula using the high-resolution Polar WRF simulations, and identify four distinct AR shapes that cause föhn-induced surface warming over the LCIS: fully zonal, zonal-like, concave, and convex. They first analyze statistical results related to the landfall positions, pressure fields, IVT, and impacts on temperature and radiative fluxes for each AR type. And then they conduct detailed case studies of two specific AR shapes—fully zonal and convex—examining changes in AR-related wind fields, shortwave and longwave radiation, and associated föhn events. Their findings indicate that convex ARs produce the strongest föhn warming, while fully zonal ARs, due to their accompanying moisture and cloud cover, suppress the increase in föhn-driven shortwave radiation, resulting in more typical warming magnitudes. In contrast, zonal-like and concave ARs exert greater influence on the northern part of the LCIS.

The two key takeaway messages of the manuscript appear to be:

- 1) ARs making landfall on the LCIS can be categorized into four distinct types: fully zonal, zonal-like, concave, and convex.
- 2) Convex ARs cause the most intense föhn warming; fully zonal ARs lead to typical increases in temperature; and zonal-like and concave ARs have a more pronounced impact on the northern LCIS.

Overall assessment:

Investigating the types of ARs making landfall on the LCIS and their respective impacts on föhn warming is an interesting and valuable topic that could significantly advance our understanding of ARs in Antarctica. The study is generally well-presented, engaging, and supported by high-quality figures. However, for a journal with the high profile of Nature Communications, the two key takeaways in their current form are somewhat limited and would benefit from being framed to convey broader implications. Although the results are intriguing and potentially important, I have several major concerns that should be thoroughly addressed. While these issues are, in principle, resolvable, the extent of additional analysis and revisions required leads me to lean toward a recommendation of "reject and resubmit," particularly if this were a discipline-specific journal.

- 1) The AR detection method requires significant clarification. The manuscript notes the use of IVT but fails to specify the algorithmic details—such as the threshold type (fixed, relative, or none), geometric constraints, and whether a single or composite method was applied. Since AR statistics are notoriously dependent on the detection methodology (as robustly

shown by ARTMIP studies), it is crucial that the authors utilize multiple algorithms and include a discussion on this uncertainty. This remains necessary even considering the potential uniqueness of polar ARs.

Shields, C. A., Payne, A. E., Shearer, E. J., Wehner, M. F., O'Brien, T. A., Rutz, J. J., Leung, L.R., Ralph, F. M., Collow, A. B. M., Ullrich, P. A. Ullrich, Dong, Q., Gershunov, A., Griffith, H., Guan, B., Lora, J. M., Lu, M., McClenny, E., Nardi, K. M., Pan, M., Qian, Y., Ramos, A. M. Ramos, Shulgina, T., Viale, M., Sarangi, C., Tomé, R., Zarzycki, C. (2023). Future atmospheric rivers and impacts on precipitation: Overview of the ARTMIP Tier 2 high-resolution global warming experiment. *Geophysical Research Letters*, 50, e2022GL102091. <https://doi.org/10.1029/2022GL102091>

Collow, A.B., Shields, C.A., Guan, B., Kim, S., Lora, J.M., McClenny, E.E., Nardi, K., Payne, A., Reid, K., Shearer, E. J. , Tome, R., Wille, J.D., Ramos, A.M., Gorodetskaya, I.V., Leung, L.R., O'Brien, T.A., Ralph, F.M., Rutz, J. Ullrich, P.A., Wehner, M., (2022) An Overview of ARTMIP's Tier 2 Reanalysis Intercomparison: Uncertainty in the Detection of Atmospheric Rivers and their Associated Precipitation, *Journal of Geophysical Research, Atmospheres*, <https://agupubs.onlinelibrary.wiley.com/doi/10.1029/2021JD036155>.

O'Brien, Travis Allen and Wehner, Michael F and Payne, Ashley E. and Shields, Christine A and Rutz, Jonathan J. and Leung, L. Ruby and Ralph, F. Martin and Marquardt Collow, Allison B. and Guan, Bin and Lora, Juan Manuel and et al., (2022) Increases in Future AR Count and Size: Overview of the ARTMIP Tier 2 CMIP5/6 Experiment. *JGR-A* <https://agupubs.onlinelibrary.wiley.com/doi/10.1029/2021JD036013>.

Rutz, J.J, Shields, C.A., Lora, J.M, Payne, A.E., Guan, B., Ullrich, P., O'Brien, T., Leung, L.-Y., Ralph, F.M., Wehner, M., Brands, S., Collow, A., Goldenson, N., Gorodetskaya, I., Griffith, H., Hagos, S., Kashinath, K., Kawzenuk, B., Krishnan, H., Kurlin, V., Lavers, D., Magnusdottir, G., Mahoney, K., McClenny, E., Muszynski, G., Nguyen, P.D., Prabhat, Qian, Y., Ramos, A.M., Sarangi, C., Sellars, S., Shulgina, T., Tome, R., Waliser, D., Walton, D., Wick, G., Wilson, A., Viale, M.: The Atmospheric River Tracking Method Intercomparison Project (ARTMIP): Quantifying Uncertainties in Atmospheric River Climatology, *Journal of Geophysical Research-Atmospheres*, <https://doi.org/10.1029/2019JD030936>, 2019.

Shields, C. A., Rutz, J. J., Leung, L.-Y., Ralph, F. M., Wehner, M., Kawzenuk, B., Lora, J. M., McClenny, E., Osborne, T., Payne, A. E., Ullrich, P., Gershunov, A., Goldenson, N., Guan, B., Qian, Y., Ramos, A. M., Sarangi, C., Sellars, S., Gorodetskaya, I., Kashinath, K., Kurlin, V., Mahoney, K., Muszynski, G., Pierce, R., Subramanian, A. C., Tome, R., Waliser, D., Walton, D., Wick, G., Wilson, A., Lavers, D., Prabhat, Collow, A., Krishnan, H., Magnusdottir, G., and Nguyen, P.: Atmospheric River Tracking Method Intercomparison Project (ARTMIP): project goals and experimental design, *Geosci. Model Dev.*, 11, 2455-2474, <https://doi.org/10.5194/gmd-11-2455-2018>, 2018.

2) The study employs high-resolution Polar WRF to simulate 37 AR-Föhn events identified from ERA5 data (2001-2022). While this approach is valuable, the limited sample size of 37 events—further subdivided into four (effectively five) distinct AR types—results in a small number of cases per category. This raises concerns regarding the statistical robustness and generalizability of the findings. To enhance the reliability of the conclusions, I recommend conducting additional supplemental PWRP simulations to increase the sample size for each AR type.

3) While the study appropriately applies the Gridded Enhanced AR Scale for classification, the potential of this framework is not fully realized. The analysis focuses on AR2 in cases but overlooks the other categories. To leverage the scale effectively, the authors should discuss the prevalence of AR1 and AR3 events (evident in Fig. 2a) within each shape category and analyze how AR intensity, combined with shape, modulates the hydrological and thermal impacts on the LCIS.

4) The Background section currently provides separate introductions to the warming effects on the Antarctic Peninsula (AP), the impacts of ARs and föhn on the ice sheet, and the influence of AR-föhn events on AP temperatures. For a journal of Nature Communications' stature, this presentation is somewhat lengthy and could be significantly streamlined to present a more focused and concise narrative. And the Summary section primarily reiterates the main findings, resulting in repetition rather than synthesis. To meet the high standards of Nature Communications, this section should be evolved into a robust Discussion. It should delve into the implications of the results, address uncertainties, and outline unresolved questions that warrant future investigation

5) The core novelty of the study lies in the classification of ARs and the subsequent analysis of the distinct atmospheric and radiative impacts of the four types. While valuable, this may not sufficiently constitute a major conceptual advance expected for a high-impact journal. To significantly strengthen the manuscript's contribution, I recommend expanding the analysis to investigate the downstream effects of these AR-föhn events on sea ice melt.

Additional specific comments:

1) Figs 1 and 2: I suggest consolidating Figure 1 and Figure 2b into a single, more comprehensive figure. Meanwhile, Figure 2a could be moved to the supplementary information.

2) Lines 185-186 and Fig.4: It would be helpful to see the spatial patterns of the latent heat flux and longwave radiation to better evaluate their roles in the surface energy budget and the resulting melting processes.

3) Lines 480-481: Please include a brief explanation of the physical significance of Fr values, specifically stating what $Fr > 1$ and $Fr < 1$ represent in the context of this study.

4) Lines 236-238 and Fig. 8: For the Fully Zonal AR type, the temporal evolution appears consistent with the patterns of IVT

and downward longwave radiation. However, the SHF exhibits a lag of approximately 6 hours. To understand this discrepancy, it would be informative to examine the profile of diabatic heating.

5) Lines 264-268 and Fig.8: For the convex AR case, the SHF does not exhibit a pronounced change and its temporal evolution does not align closely with the observed variations in 2-meter temperature.

6) Lines 283-297: These two sentences can be combined and made more concise.

7) Lines 287-289: Is SHF also a reduction?

8) Table S1: suggested to add the AR type.

9) Table S2: Lack of "fully meridional"

10) Data: While the model performance is convincingly demonstrated for two representative cases, this validation would be strengthened by a more comprehensive evaluation. Could the authors please present statistical metrics (e.g., mean bias, correlation) that aggregate the results across all 37 simulated AR-föhn events?

Version 1:

Reviewer comments:

Reviewer #1

(Remarks to the Author)

The authors have effectively addressed all previously raised concerns and suggestions. They have thoroughly revised the manuscript, incorporating the necessary changes and improvements. I appreciate the diligence and clarity demonstrated in the revisions. I have no further questions or concerns regarding the content, significance, or quality of the paper. In my opinion, the manuscript now meets the required standards of the journal, and I recommend it for publication.

Reviewer #2

(Remarks to the Author)

All of my concerns have been addressed by the authors.
I just have comments about a few grammatical errors in the sentences.

L109 : Table. S1 -> Table S1

L194, L212 : exhibites -> exhibits

L248 : approach -> approached

L338 : shows -> show

Reviewer #3

(Remarks to the Author)

Thank you for your efforts in addressing all my initial comments. I am satisfied with most of the revisions. However, I still have a few remaining concerns and suggestions, as outlined below:

(1) The definition and selection criteria for "AR-föhn events" could be more clearly articulated. I suggest providing a more detailed description of the identification process in the Methods section, including specific thresholds, time windows, and spatial extents, to enhance reproducibility and facilitate comparison with future studies.

(2) While the cluster analysis successfully identified four distinct AR shapes, the authors also note that some shapes (e.g., zonal-perpendicular and concave) are not always clearly distinguishable. I recommend including a discussion on this limitation and, if possible, exploring complementary classification methods or diagnostics to further validate the grouping.

(3) Regarding the AR scale classification, it would be helpful to explicitly highlight in the main text the limitations associated with AR1 and AR3 events, and to indicate whether future work will specifically address AR3 events in more detail.

(4) Please ensure that all abbreviations appearing in figures and tables are spelled out in full at their first occurrence, in accordance with standard journal guidelines.

REVIEWER COMMENTS

Reviewer #1 (Remarks to the Author):

This study explores the relationship between atmospheric river (AR) characteristics and foehn-induced surface warming over the Larsen C Ice Shelf (LCIS) in Antarctic Peninsula (AP) using a gridded AR dataset and high-resolution Polar-WRF model simulations for each AR event over 2001-2022. Four distinct AR shapes are identified, and respective physical mechanisms are proposed to explain their impact on foehn warming. The results are novel and significant, which should be of interest to the readership of Nature Communications. The findings have great potential implications for Earth system modeling of Antarctic ice shelf stability and loss. However, there are some technical concerns related to the methodology and analysis. I cannot recommend it for publication in the current form. The following specific comments need to be satisfactorily addressed.

Thank you very much for the valuable comments. We have addressed all of Reviewer #1's suggestions and provide point-by-point responses below.

1) When referring to AR shape and landfall location, I would picture the geometric shape of long, narrow moisture corridor upon landfalling over AP, but the schematic illustration in Figure 2 got me confused. Not sure whether the AR pathway represents the moving path of AR. How is the AR pathway for each event identified from the ERA5 data and drawn in Fig. 2a? What do the dates and the length of each line mean?

We examine IVT fields at 6-hourly intervals and manually identify the dominant AR pattern associated with each event (we replaced the previous terminology "AR pathway", which was confusing). These results are shown in Fig. S1 (originally Fig. 2a).

Taking the 7 February 2022 event as an example, the AR initially exhibits a concave shape, rapidly transitions to a zonal-like shape, and eventually evolves into a fully meridional shape. The dominant AR shape during this event is zonal-like, with peak impact occurring around 00 UTC on 8 Feb 2022. Accordingly, we represent the dominant AR configuration using the orange arrow (panel e; see also Fig. 1c, originally Fig. 2a). We include more explanation in the Supplementary Materials.

REMA Topography

“**Supplementary Figure S1.** Dominant AR shape for each AR-föhn event impacting the AP during austral summers from 2001 to 2022, as well as corresponding dates and ranking on the Enhanced AR Scale¹. AR shapes of AR-family events are shown by dashed lines and a “F” after the date. If an AR-family event includes multiple shapes, the secondary shape is indicated by a dashed line. Also shown is the model topography (m) from D02 at 6 km, based on the REMA dataset. LCIS in (b) refers to the Larsen C Ice Shelf. The dominant AR shape is manually defined based on the IVT field, following the AR axis at the peak of each event.”

This method has been widely used in studies of California ARs and effectively captures the overall AR event behavior (see figure below), which we reference.

2) PWRf configuration: How are the model domains aligned with the zonal and meridional bands near AP?

Thank you for pointing this out. Our domain configuration follows a standard regional climate model setup for the Antarctic Peninsula and will provide reliable AR simulations. Further details are provided below.

There are a couple of related concerns:

a. When nudging u and v winds in the PWRf simulations to 6-hourly ERA5, should the ERA5 grids be remapped and spatially interpolated to the PWRf x and y directions? How?

Yes—interpolation is applied to convert the ERA5 fields to the model grid so that nudging can be performed during the simulations.

As the reviewer probably knows, the WRF workflow consists of five steps: *geogrid*, *ungrib*, *metgrid*, *real*, and *wrf*. Although PWRf incorporates several modifications to physical parameterizations, it still follows this standard WRF workflow.

During the *metgrid* step, the meteorological fields extracted by *ungrib* are horizontally interpolated to the model grid defined in *geogrid*¹. In the subsequent *wrf* step, the PWRf simulation is nudged toward ERA5 fields (intermediate data generated by *metgrid*) to

¹ https://www2.mmm.ucar.edu/wrf/users/wrf_users_guide/build/html/wps.html).

reduce model degradation. The nudging configuration used in our study is typical for weak-to-moderate synoptic-scale constraint.

This approach has been tested in prior studies and is recommended by the PWRF development team. We apologize for reiterating model details the reviewer may already be familiar with, and we hope this explanation adequately addresses the question.

To maintain conciseness while avoiding confusion, we now include brief explanations of this procedure in both the main text and Table S3.

Ln 429-430: “PWRF follows the standard WRF workflow but incorporates multiple updates to its physical parameterizations.”

Supplementary Table S3: “Every 6 hours; nudging to *u*, *v* wind, temperature and water vapor from ERA5 fields (preprocessed and interpolated after the metgrid step)^{8,9} for model level 40 (~400 hPa) and above...”

b. Please clarify how the domain alignments may affect the PWRF simulated AR pathways and shape patterns that are defined relative to the zonal lines. The Fully Zonal AR shape/pathway depicted in Fig. 2b doesn't align with the zonal lines (gray dashed lines).

The term “*fully zonal*” is ambiguous in the polar context, as zonal flow implies circulation around the Antarctic continent. We have changed it to “***zonal-perpendicular***” to emphasize that the flow is oriented perpendicular to the mountain range upstream of the LCIS, thereby promoting strong cross-barrier airflow.

In addition, our domain configuration follows a fairly standard regional climate model setup for the Antarctic Peninsula (AP; e.g., Listowski and Lachlan-Cope 2017, Luo et al. 2020). This design keeps the AP near the center of the domain, ensuring that key regions such as Larsen C are not influenced by lateral boundary effects.

Our simulation results are not sensitive to the PWRF domain configuration because:

1. D01 and D02 domains are sufficiently large to include most of the Southern Ocean.
2. Domain boundaries are chosen to minimize topographic truncation, thereby reducing the risk of model instability (although some truncation of the terrain is unavoidable).

- Listowski, C. and Lachlan-Cope, T. (2017): The microphysics of clouds over the Antarctic Peninsula – Part 2: modelling aspects within Polar WRF, *Atmos. Chem. Phys.*, 17, 10195–10221, <https://doi.org/10.5194/acp-17-10195-2017>.
- Luo, L., Zhang, J., Hock, R., & Yao, Y. (2021): Case study of blowing snow impacts on the Antarctic Peninsula lower atmosphere and surface simulated with a snow/ice enhanced WRF model. *J. Geophys. Res. Atmos.*, 126, e2020JD033936. <https://doi.org/10.1029/2020JD033936>

3) Figure 1: embedding the D01 at the corner of D02, with the shared axes, is very confusing. I suggest moving it out of D02 and marking its own latitude-longitude ranges.

Thank you for the suggestion. We have merged this comment with a related comment from another reviewer. Accordingly, Figure 1 has been updated, and Figure 2a has been moved to the Supplementary Materials as Fig. S1.

“Fig. 1. PWRF model domains and overview of four AR shapes: a) D01 with a horizontal resolution of 30 km; b) D02 and D03 with resolutions of 6 km and 1.2 km, respectively; c) Classification of all events shown in Supplementary Fig. S1 into distinct AR shapes, which are zonal-perpendicular, zonal-like, convex, and concave. In panel b, the purple dashed line marks the AR detection zone; Spring Point, AWS 14, AWS 15 and AWS 17 are shown as red crosses, and the Escudero station as a purple dot. Topography in panels b-c is from the 6-km WRF D02 domain based on Reference Elevation Model of Antarctica (REMA). Larsen C denotes the Larsen C Ice Shelf.”

4) The Froude (Fr) number, being used to depict the AP topographic flow blocking, might be able to objectively quantify the impact of AR shape and landfall location on foehn-induced warming. Why was Fr number arbitrarily calculated using the mean U calculated in the small blue box? How was the elevation range 900 m – 1100 m identified?

Thank you for this suggestion. After careful discussion among all authors, we decided to remove the Froude number calculation to avoid potential confusion, as the trajectory analysis and cross sections already clearly indicate whether low-level blocking occurs on the upwind side.

To answer the reviewer’s question, the small blue box was selected because (1) it is sufficiently far from the Domain 3 boundary to avoid potential model instability, and (2) it lies more than ~200 km from the mountain peak—beyond the typical Rossby radius of deformation. The average elevation is defined based on the mountain peak height along the AP. Height is estimated based on the AR landfall location and the corresponding mountain elevation.

5) I tend to believe that the AR landfalling location and wind speed (near the height of the topographic elevation range) might be more important to foehn-induced warming than the specific AR patterns (identified by the IVT) and moisture content. Taking the convex AR (2013-02-23), if most of the AR moisture is sourced from distant ocean along an elevated pathway, it’s

hardly convincing that the AR moisture can have a more significant impact on the LCIS through SHF and downward SW processes than the downward LW radiation. Please comment on this.

Landfall location & Wind speed VS. AR shape.

Yes, we agree with the reviewer that landfall location and wind speed are indeed very important factors, together with wind direction, airflow curvature, and the vertical position of the AR core, as these collectively determine whether the flow can traverse the mountain barrier and the magnitude of the resulting föhn warming. We believe that **AR shape** encapsulates these key features to a certain extent and also links to the larger-scale circulation patterns.

We added the following discussion to the Discussion section regarding this comment. Please check Ln 358-376.

SEB analysis for the convex AR

We believe that our findings are in line with the reviewer's perspective; however, this was not clearly articulated in the original manuscript.

Downward LW radiation can exceed the impact of downward SW, typically where AR-associated moisture exerts the strongest influence (e.g., the 2013 convex event). The contribution from downward LW radiation is more consistent, whereas downward SW radiation exhibits a pronounced diurnal cycle. However, downward SW radiation has a much higher absolute value during the daytime. Please see the detailed response below and the corresponding revisions in the manuscript.

When we look at the LCIS as a whole.

Following panels are selected from Fig. 7.

1. Downward SW typically exhibits larger magnitudes than other SEB components over the Antarctic Peninsula during austral summer, featuring a strong diurnal cycle. In the convex AR case, although downward SW remains the dominant SEB component, it shows a weakening trend over the course of the event.
2. By contrast, downward LW increases steadily during the event; however, it does not become the dominant component on average because its absolute contribution is generally smaller than that of downward SW.
3. SHF displays strong spatial variability and doesn't have the diurnal cycle. Its maximum values can locally exceed those of both downward SW/LW.

When we look at the spatial distribution of SEB over the LCIS.

As the reviewer pointed out, in regions where AR-induced moisture exerts the strongest influence—such as the southern LCIS during the convex event—downward LW can temporarily exert a stronger influence than downward SW.

Ln 293-296: *“Because SHF exhibited strong spatial variability and affected only limited areas, its median value remained relatively stable during the event. Over the southern LCIS, where AR-associated moisture exerted the strongest influence, enhanced downward LW could exceed the contribution from downward SW (not shown).”*

We added the following content to the Discussion section regarding this comment.

Ln 342-344: *“Increased surface downward SW radiation along the LCIS margin is consistent with föhn-induced cloud clearing, while enhanced AR-related moisture and downward LW radiation near mountain gaps (Fig. 3) can locally exceed the SW contribution.”*

We also highlighted the importance of downward LW radiation, for example:

Ln 189-194: *“Consistent with previous studies showing that isentropic drawdown, mechanical mixing, and radiative heating are key processes driving föhn-induced warming over the AP¹⁸, sensible heat flux (SHF) together with downward SW and downward longwave (LW) radiation dominate the SEB over the LCIS (Fig. 3)^{14,20,33}, i.e., more important than other components like latent heat flux (Supplementary Fig. S2a-d).”*

6) Reanalysis products can be much biased over polar regions. I wonder if the authors tried a different reanalysis product than ERA5 for driving the PWRP model.

Thank you for this insightful comment. Indeed, reanalysis products can exhibit substantial biases over polar regions. Among currently available datasets, ERA5 is considered one of the most reliable for Antarctica, particularly over the Antarctic Peninsula. As shown by Bromwich et al. (2024), ERA5 temperature trends closely match observations at the Antarctic Peninsula stations Faraday/Vernadsky, Esperanza, and Orcadas. In addition, ERA5 has been widely used for dynamic downscaling studies in

polar regions (e.g., Bozkurt et al., 2018; Xue et al., 2022; Zou et al., 2023), providing a robust foundation for high-resolution simulations.

This study generated approximately 40 TB of output after cleanup (nearly 100 TB of data during processing) and consumed about three million RUs. Nevertheless, we appreciate the reviewer's suggestion and agree that a systematic evaluation of different reanalysis products for AR studies would be a valuable direction for future research. We have therefore added the following context in the Data section.

Ln 471-473: *“Future research will help further improve PWRP simulations by exploring alternative reanalysis datasets for initial and boundary conditions, refining model physical parameterizations, and incorporating additional observations.”*

Reviewer #2 (Remarks to the Author):

This study selected cases of Atmospheric Rivers (ARs) that could accompany föhn phenomena in the Larsen C Ice Shelf (LCIS) region from 2001 to 2022 in the Antarctic Peninsula (AP), and investigated the föhn mechanisms based on the trajectory shapes of the ARs. For detailed analysis of these mechanisms, high-resolution downscaling data (1.2 km) were generated using Polar WRF. Using this dataset, the ARs were classified by shape, and the warming patterns in the AP region were examined. The föhn mechanisms were analyzed in detail for two representative AR cases.

Given the recent rapid increase in the frequency of föhn-induced warming events on the Antarctic Peninsula due to accelerated climate change, this study aims to significantly contribute to understanding and addressing Antarctic warming.

Thank you very much for the valuable comments. We have addressed all of Reviewer #2's suggestions and provide point-by-point responses below.

Major Concern

1) This study classified AR shapes using K-means clustering, examined the characteristics of each cluster, and then analyzed the mechanisms of föhn impacts through two detailed AR case studies. The analysis of the föhn mechanisms in these cases is a very important part of the research, as it quantitatively examines surface warming depending on the föhn processes. However, the reason for selecting the fully zonal and convex-shaped ARs, as well as the rationale for choosing the two corresponding cases, is insufficient. It would be necessary to select the target shapes based on at least some quantitative evidence, such as the frequency of each AR or a quantitative comparison of the intensity of warming over LCIS. Similarly, in case selection, providing quantitative selection criteria such as LCIS temperature thresholds would strengthen the justification.

Thank you for pointing this out. Following the reviewer's suggestion, we refined our analysis in a more quantitative manner and added a detailed, step-by-step description of the workflow to clarify the methodology. The "*Supplementary Material: Selection Criteria and Analysis Method for AR Events and Föhn-Warming Periods*" now includes a comprehensive description of the procedures used to identify AR events, define föhn-warming periods, perform k-means clustering analysis, and select the two representative cases.

Below we outline the key questions addressed in this response:

- How does föhn warming period decide?
For each AR event, we plotted a time series of the maximum IVT and the maximum 2m temperature over the LCIS only (from PWRP simulations). To objectively identify föhn-induced warming, the diurnal cycle was removed from the temperature time series (solid red), and warming periods were selected only if they persisted for more than 6 hours and exhibited a temperature increase exceeding 3 °C (highlighted by the pink shading).

- Why select fully-zonal and convex?

Convex. Although relatively rare, this AR shape produces persistent warming across the entire LCIS, particularly over the southern part.

Zonal-perpendicular (originally fully zonal): This pattern is selected due to our hypothesis that near-perpendicular flow impinging on the mountain range leads to stronger föhn warming.

Zonal-like: This is a well-studied AR shape. Previous studies (e.g., Zou et al. 2023; Gorodetskaya et al. 2024) have comprehensively examined the February 2022 zonal-like event.

Concave: The AR-induced föhn warming response associated with this pattern is less consistent and on average weak. We frequently observe barrier-jet features in these cases, and further analysis might be needed.

Ln 180-185: *“Despite having lower IVT intensity than concave and zonal-like ARs, the convex AR shape still favors strong warming over the southern sector (Fig. 2), highlighting the unique impact of this AR shape (see the case study below). The concave AR shape, however, produces the weakest föhn warming, confined to the northern LCIS near the ice-shelf edge. This is likely owing to its landfall location and the curved geometry of the AP, which enhance mountain blocking and thereby limit the extent of föhn warming.”*

- Why those two cases?

The 2013 convex event was selected first because it ranked third in föhn warming among all events (8.4 °C increase in T2_{max}; see figure above) and ranked second when AR-family events were excluded (Table S1). This also motivated our focus on the convex AR shape.

The 2008 zonal-perpendicular event was selected subsequently based on three criteria:

- It is a 3-day AR2 event, comparable in duration & intensity to the convex case.
- It is a relatively strong föhn-warming event.
- It exerts a clear warming impact over the central and southern LCIS.

**Other strong zonal-perpendicular events not selected: Case 18 (2011) is an AR3 event, and Case 32 (2021) primarily affects the northern LCIS.*

In addition, AR-family events are excluded from the detailed analysis to focus on cases dominated by a single AR shape, thereby providing a clearer assessment of the impact. Table S1 now provides additional information supporting our case-selection decisions.

Ln 236-242: *“The zonal-perpendicular AR-föhn event occurred from 1-3 Dec 2008 and the convex AR-föhn event from 22-24 Feb 2013. Notably, the 2013 convex event exhibited the strongest föhn warming over the LCIS among all convex-dominated AR events ($T_{2_{max}}$ increased by 8.4 °C; Table S1) and ranked third among all events considered. The 2008 zonal-perpendicular event represents a strong föhn warming within its shape category ($T_{2_{max}}$ increased by 4.4 °C; Table S1). Hereafter, these cases are referred to as the “zonal-perpendicular” and “convex” events, respectively.”*

Minor Comments

L40: thereby have -> thereby having a
Corrected

L116: shapes -> Shapes
Corrected

L120: Why was the analysis period set from 2001 to 2022? Since ERA5 data cover a longer period, and reanalysis-based studies typically begin from 1979, the reason for choosing 2001 as the starting year should be clarified.

Yes, it is because of the availability of MODIS albedo data.

In Method section, we added an explanation as the reviewer suggested:

Ln 484-485: *“This study restricts event selection to years after 2001 because our high-resolution simulations incorporate MODIS albedo observations, which are only available after February 2000.”*

L273: 9 hour -> 9-hour
Corrected

L308: determines -> determine
Corrected

In the methods section, a more detailed description of the K-means clustering approach is needed. Was the number of k predetermined?

Yes, the number of k is predetermined as 4, after some pre-analysis (3-5). Also, it better captures all patterns we observed in the 6hrly IVT and 500hPa geopotential height figures we generated (archived at AMRDC).

- Zou, X. et al. Selected austral summer Atmospheric River cases over the Antarctic Peninsula, 2001 - 2022. Antarctic Meteorological Research and Data Center <https://doi.org/10.48567/exak-2810> (2025).

In the *Supplementary Material: Selection Criteria and Analysis Method for AR Events and Föhn-Warming Periods*

“K-means clustering analysis uses the Hartigan and Wong (AS-136) algorithm¹⁰, which partitions multidimensional data points into K clusters by minimizing within-cluster variance. It was applied separately to 500-hPa geopotential height and IVT for all 1,165 selected hours of the PWRP D02 simulations and for additional pre-2001 events from the ERA5 reanalysis dataset (not shown). The number of clusters ($k = 4$) was determined based on manual inspection of all 6-hourly IVT fields, which indicated that four distinct AR shapes occur during AR-induced föhn-warming periods.”

The term “fully zonal” seems clear in the context of shape classification, but since the trajectories are not strictly parallel to latitude, the use of “fully” may not be linguistically precise.

Thank you for the suggestion. Another reviewer raised a similar concern; thus, we changed it to “**zonal-perpendicular**” now.

In k-means cluster analysis, typical information provided in other clustering studies—such as the number of ARs per cluster, mean duration, IVT, and degree of warming—appears to be missing.

Thank you very much for this valuable suggestion. We re-performed the analysis:

- We applied a more quantitative event-selection procedure (see Supplementary Materials)
- We examined surface impacts by compositing surface variables (T2, SHF, and SWD, LWD, LH, LWnet, SWnet) according to the IVT-based clusters.

We added the percentages to indicate the fraction of time steps assigned to each cluster. Because a given AR event evolves over its life cycle, a single AR case can contribute to multiple clusters at different stages of development.

All related analyses and conclusions have been revised accordingly. Please check Ln 197-234.

Surface variables (T2, SHF, SWD, and LWD) were then composited according to the IVT-based clusters, and mean values were calculated for each cluster. We removed the diurnal cycle from T2 and calculated the hourly rate of the change ($dT2/dt$).

Reviewer #3 (Remarks to the Author):

Summary :

In "Föhn-Induced Melting over Larsen C Modulated by Atmospheric River Shape, Direction and Landfall Location," Zou et al. analyze the relationship between ARs and föhn over the Antarctic Peninsula using the high-resolution Polar WRF simulations, and identify four distinct AR shapes that cause föhn-induced surface warming over the LCIS: fully zonal, zonal-like, concave, and convex. They first analyze statistical results related to the landfall positions, pressure fields, IVT, and impacts on temperature and radiative fluxes for each AR type. And then they conduct detailed case studies of two specific AR shapes—fully zonal and convex—examining changes in AR-related wind fields, shortwave and longwave radiation, and associated föhn events. Their findings indicate that convex ARs produce the strongest föhn warming, while fully zonal ARs, due to their accompanying moisture and cloud cover, suppress the increase in föhn-driven shortwave radiation, resulting in more typical warming magnitudes. In contrast, zonal-like and concave ARs exert greater influence on the northern part of the LCIS.

The two key takeaway messages of the manuscript appear to be:

- 1) ARs making landfall on the LCIS can be categorized into four distinct types: fully zonal, zonal-like, concave, and convex.
- 2) Convex ARs cause the most intense föhn warming; fully zonal ARs lead to typical increases in temperature; and zonal-like and concave ARs have a more pronounced impact on the northern LCIS.

Thank you very much for the valuable suggestions. We have provided point-by-point responses to all of Reviewer #3's comments.

We also note that some of our conclusions have been updated following the application of a stricter selection criterion (more quantitative) for identifying föhn warming and the adoption of a more robust analysis procedure (see Supplementary Material).

Overall assessment:

Investigating the types of ARs making landfall on the LCIS and their respective impacts on föhn warming is an interesting and valuable topic that could significantly advance our understanding of ARs in Antarctica. The study is generally well-presented, engaging, and supported by high-quality figures. However, for a journal with the high profile of Nature Communications, the two key takeaways in their current form are somewhat limited and would benefit from being framed to convey broader implications. Although the results are intriguing and potentially important, I have several major concerns that should be thoroughly addressed. While these issues are, in principle, resolvable, the extent of additional analysis and revisions required leads me to lean toward a recommendation of "reject and resubmit," particularly if this were a discipline-specific journal.

Thank you for pointing this out. We agree that broader implications are important for a journal such as Nature Communications, and we have added the following materials to strengthen the broader impact of the paper:

- 1) **Background:** We trimmed back this section to focus on key messages.

- 2) **Results:** We refined the AR–föhn event selection algorithm, improved the k-means clustering framework, and extended some analysis beyond 2001 using ERA5 reanalysis data. These additions help assess the representativeness of our event subset and suggest that incorporating additional AR characteristics may have broader relevance for future Antarctic AR studies.
- 3) **Discussion:** This section has been completely rewritten to incorporate all key points raised by the reviewers.

1) The AR detection method requires significant clarification. The manuscript notes the use of IVT but fails to specify the algorithmic details—such as the threshold type (fixed, relative, or none), geometric constraints, and whether a single or composite method was applied. Since AR statistics are notoriously dependent on the detection methodology (as robustly shown by ARTMIP studies), it is crucial that the authors utilize multiple algorithms and include a discussion on this uncertainty. This remains necessary even considering the potential uniqueness of polar ARs.

Shields, C. A., Payne, A. E., Shearer, E. J., Wehner, M. F., O'Brien, T. A., Rutz, J. J., Leung, L.R., Ralph, F. M., Collow, A. B. M., Ullrich, P. A. Ullrich, Dong, Q., Gershunov, A., Griffith, H., Guan, B., Lora, J. M., Lu, M., McClenny, E., Nardi, K. M., Pan, M., Qian, Y., Ramos, A. M. Ramos, Shulgina, T., Viale, M., Sarangi, C., Tomé, R., Zarzycki, C. (2023). Future atmospheric rivers and impacts on precipitation: Overview of the ARTMIP Tier 2 high-resolution global warming experiment. *Geophysical Research Letters*, 50, e2022GL102091.
<https://doi.org/10.1029/2022GL102091>

Collow, A.B., Shields, C.A., Guan, B., Kim, S., Lora, J.M., McClenny, E.E., Nardi, K., Payne, A., Reid, K., Shearer, E. J. , Tome, R., Wille, J.D., Ramos, A.M., Gorodetskaya, I.V., Leung, L.R., O'Brien, T.A., Ralph, F.M., Rutz, J. Ullrich, P.A., Wehner, M., (2022) An Overview of ARTMIP's Tier 2 Reanalysis Intercomparison: Uncertainty in the Detection of Atmospheric Rivers and their Associated Precipitation, *Journal of Geophysical Research, Atmospheres*,
<https://agupubs.onlinelibrary.wiley.com/doi/10.1029/2021JD036155>.

O'Brien, Travis Allen and Wehner, Michael F and Payne, Ashley E. and Shields, Christine A and Rutz, Jonathan J. and Leung, L. Ruby and Ralph, F. Martin and Marquardt Collow, Allison B. and Guan, Bin and Lora, Juan Manuel and et al., (2022) Increases in Future AR Count and Size: Overview of the ARTMIP Tier 2 CMIP5/6 Experiment. *JGR-A*
<https://agupubs.onlinelibrary.wiley.com/doi/10.1029/2021JD036013>.

Rutz, J.J, Shields, C.A., Lora, J.M, Payne, A.E., Guan, B., Ullrich, P., O'Brien, T., Leung, L.-Y., Ralph, F.M., Wehner, M., Brands, S., Collow, A., Goldenson, N., Gorodetskaya, I., Griffith, H., Hagos, S., Kashinath, K., Kawzenuk, B., Krishnan, H., Kurlin, V., Lavers, D., Magnusdottir, G., Mahoney, K., McClenny, E., Muszynski, G., Nguyen, P.D., Prabhat, Qian, Y., Ramos, A.M., Sarangi, C., Sellars, S., Shulgina, T., Tome, R., Waliser, D., Walton, D., Wick, G., Wilson, A., Viale, M.: The Atmospheric River Tracking Method Intercomparison Project (ARTMIP): Quantifying Uncertainties in Atmospheric River Climatology, *Journal of Geophysical Research-Atmospheres*, <https://doi.org/10.1029/2019JD030936>, 2019.

Shields, C. A., Rutz, J. J., Leung, L.-Y., Ralph, F. M., Wehner, M., Kawzenuk, B., Lora, J. M., McClenny, E., Osborne, T., Payne, A. E., Ullrich, P., Gershunov, A., Goldenson, N., Guan, B., Qian, Y., Ramos, A. M., Sarangi, C., Sellars, S., Gorodetskaya, I., Kashinath, K., Kurlin, V., Mahoney, K., Muszynski, G., Pierce, R., Subramanian, A. C., Tome, R., Waliser, D., Walton, D., Wick, G., Wilson, A., Lavers, D., Prabhat, Collow, A., Krishnan, H., Magnusdottir, G., and Nguyen, P.: Atmospheric River Tracking Method Intercomparison Project (ARTMIP): project goals and experimental design, *Geosci. Model Dev.*, 11, 2455-2474, <https://doi.org/10.5194/gmd-11-2455-2018>, 2018.

Thank you for pointing this out. Uncertainty in AR detection is important and remains an active area of research.

In this study, we focus on impactful ARs over the AP, a region characterized by lower water vapor saturation capacity, where both meridional and zonal moisture transport are important. Our detection method is based on the Enhanced AR Scale, which was widely used during the Year of Polar Prediction (YOPP) Southern Hemisphere Targeting Observation Period and has provided reliable estimates of AR impacts. Thus, we believe our AR selection results are robust for this study.

To address reviewer’s comments, we completed the following tasks:

- 1) Added a detailed section in the Supplementary: *Selection Criteria for AR Events and Föhn-Warming Periods*, to fully explain how we select all impactful AR-föhn events.

We also cross-checked all our selection against another independent dataset, Guan and Waliser (2024), to confirm its reliability. Validation figures for all cases are currently archived at AMRDC².

Case #35 as an example

²<https://amrdcdata.ssec.wisc.edu/dataset/selected-austral-summer-atmospheric-river-cases-over-antarctic-peninsula-2001-2022>

AR detection is consistent between the two datasets. Although some uncertainty exists in the spatial extent of AR impacts, this does not affect our AR identification. In other words, all selected AR events in this study are confirmed to be impactful events of at least AR1 intensity over the Antarctic Peninsula.

- Guan, B., Waliser, D.E. A regionally refined quarter-degree global atmospheric rivers database based on ERA5. *Sci Data* 11, 440 (2024). <https://doi.org/10.1038/s41597-024-03258-4>
- Guan, B. [Data] Global Atmospheric Rivers Database, Version 4. UCLA Dataverse <https://doi.org/10.25346/S6/ZSW7UN> (2025).

- 2) We added more context about uncertainties in AR detection in the Discussion section, which include all suggested citations here. Also, several authors of this study are currently contributing to a separate manuscript specifically targeting uncertainties in polar AR detection (Different metrics, spatial variations, AR families, etc). That effort, we believe, will provide a more comprehensive assessment.

Please check Ln 371-379.

2) The study employs high-resolution Polar WRF to simulate 37 AR-Föhn events identified from ERA5 data (2001-2022). While this approach is valuable, the limited sample size of 37 events—further subdivided into four (effectively five) distinct AR types—results in a small number of cases per category. This raises concerns regarding the statistical robustness and generalizability of the findings. To enhance the reliability of the conclusions, I recommend conducting additional supplemental PWRF simulations to increase the sample size for each AR type.

We appreciate this comment and would like to provide additional context that may have been missing in the original manuscript.

To properly address this comment, we completed the following tasks:

1. Expanded the analysis using ERA5 reanalysis data (1979-2001), demonstrating that our conclusion of four distinct AR shapes is robust.
2. Explained why we believe the 37 cases are sufficient to represent all AR shapes, noting that a single AR event may exhibit multiple shapes.
3. Demonstrated that all available austral-summer cases after 2001 are included in this study.

If we understand correctly, Reviewer #3 may have the following detailed concerns, all of which are important to address.

- **Do four distinct AR patterns persist in the extended dataset?**
Yes. We extended a similar IVT k-means clustering analysis for selected AR events based on ERA5 reanalysis data to show that 4 AR shapes still present.

Differences in AR shapes (e.g., landfall location) and their occurrence percentages likely arise from several factors:

- Different spatial resolutions: PWRP D02 has 6-km horizontal resolution with 71 vertical levels, whereas ERA5 has ~31-km horizontal resolution with 38 pressure levels.
- Different grid systems and map projections (global vs RCM grid).
- Our PWRP sample is additionally filtered to include only time steps with föhn warming detected over the LCIS (the detailed workflow is now included in the Supplementary).

**We did not apply an identical T2 filter to ERA5 because it is known to underestimate föhn-related warming (Gorodetskaya et al. 2023), which would bias the event selection. Thus, the ERA5 K-means clustering includes time steps that do not affect the LCIS, typically ARs making landfall farther north.*

Despite these limitations, both datasets still exhibit broadly consistent AR patterns, suggesting the key synoptic regimes are robust.

Please check Ln 379-385

- **Further proof of the representativeness of our PWRP dataset.** Individual AR events can exhibit multiple AR patterns, and although our analysis includes 37 AR events (over 3,000 hours of simulations), the dataset encompasses a sufficiently broad range of AR patterns to support robust conclusions. Several representative examples are shown here.

This is a February 1987 AR family event characterized by two back-to-back ARs based on ERA5. AR #1 initially exhibits a zonal-like structure (panel a), and then rapidly deforms into a convex AR (panel b) as the large-scale circulation evolves. This behavior is commonly observed in our analysis and motivated our decision to adopt a K-means clustering approach. AR #2 (panel c) maintains a concave structure throughout the remainder of the event. Overall, this case captures three distinct AR patterns.

This is a 7-day February 2006 AR family event characterized by three back-to-back ARs in the PWRP simulation. The event encompasses four AR patterns, excluding the convex one. We present snapshots of the different AR patterns together with the corresponding MSLP fields.

In the updated K-means clustering analysis, percentages denote the fraction of time steps assigned to each cluster; all clusters contain more than 100 time steps (See Fig. 2).

- **Have we exhausted all available AR events over the Antarctic Peninsula region? Yes.** For the Antarctic region, ARs are far less frequent than in the mid- and low latitudes; in other words, the number of naturally occurring impactful ARs is inherently limited. Wille et al. (2025), using meridional IVT alone, show that ARs typically occur for only about **3 days per year** at a given Antarctic coastal location, with notable spatial variability (left panel in the figure below). Zhang et al. (2024), which accounts for both meridional and zonal IVT, similarly finds an average of **~3 ARs exceeding AR1 intensity per year** at a typical coastal site (right panel in the figure below).

a 1980–2020 Climatology

We restrict simulations to events after 2001 because MODIS albedo observations, required for our high-resolution runs, are only available after February 2000. Thus, we are confident that our dataset includes all relatively strong AR events (AR1 or stronger) over the AP during the 2001-2022 austral summers.

We also note that this study simulates all available austral summer AR events of at least AR1 intensity using high-resolution modeling, producing approximately 40 TB of output after cleanup (nearly 100 TB of raw data generated during processing). In total, about three million RUs were used to generate this dataset. These simulations are highly computationally intensive due to the high spatial resolution, long integration periods, and comprehensive variable set.

The focus of this study is not to exhaustively simulate all AR events—particularly weaker events, which tend to have smaller impacts and are more sensitive to the choice of AR selection criteria. This study aims to provide a sufficiently representative dataset to highlight the importance of AR characteristics, which we believe is achieved in the revised manuscript.

Related contexts are revised.

3) While the study appropriately applies the Gridded Enhanced AR Scale for classification, the potential of this framework is not fully realized. The analysis focuses on AR2 in cases but overlooks the other categories. To leverage the scale effectively, the authors should discuss the prevalence of AR1 and AR3 events (evident in Fig. 2a) within each shape category and analyze how AR intensity, combined with shape, modulates the hydrological and thermal impacts on the LCIS.

Thank you for this great suggestion. Our emphasis on AR2 events is motivated by the following considerations:

- Among the 10 AR1 events, only 3 exhibit detectable föhn warming over the LCIS.
- Six AR3 events are identified, half of which are prolonged AR-family events; these complex cases warrant dedicated investigation and are therefore deferred to future work.
- The remaining 22 events are AR2 events, which form the core of our analysis.

To better address the reviewer's comment, we now explicitly include the strength of föhn warming stratified by AR scale category and expand the Discussion section to address "how AR intensity, in combination with AR shape, modulates the hydrological and thermal impacts on the LCIS."

Ln 130-138: "Among the 37 AR events, 23 events exhibited identifiable föhn-induced warming periods over the LCIS (comprising 1165 h of PWRP simulations; hereafter referred to as "AR-föhn events"), including three AR1, fourteen AR2, and six AR3 events (the latter including three prolonged AR-family events). Not surprisingly, the AR-föhn events associated with föhn-induced warming over the LCIS tended to make landfall over the central or southern AP (Supplementary Fig. S1). The average warming over the LCIS associated with the 23 AR-föhn events, defined as the increase in hourly maximum 2 m temperature ($T_{2_{max}}$) after removal of the diurnal cycle, varies from around 4.1 °C for AR1 events to 5.6 °C for AR2 events and 7.1 °C for AR3 events (Supplementary Table S1)."

Supplementary Table S1. List of AR events impacting the AP during austral summers since December 2001⁴, including their start and end dates, duration, AR scale, dominant AR shape, and the increase in hourly maximum 2 m temperature ($T_{2_{max}}$) over the LCIS after removal of the diurnal cycle. Only warming periods lasting longer than 6 hours with $T_{2_{max}}$ increases exceeding 3 °C over the LCIS are identified as föhn warming. "None" in the $T_{2_{max}}$ column indicates that no föhn warming was detected. Asterisks (*) indicate AR-family events. The AR Scale denotes the maximum AR category detected on the upwind side of the LCIS, with a rating of 1 to 3 indicating moderate to strong AR events for Antarctica as suggested by previous research⁵⁻⁷. Bold text highlights the two cases involving zonal-perpendicular and convex AR shapes selected for detailed analysis in this paper.

No.	Start Date	End Date	Duration (days)	AR Scale (ERA5)	Dominant AR shape	$T_{2_{max}}$ (°C)
1	2001-12-12	2001-12-15	4	AR2	Zonal-like	4.6
2	2002-01-01	2002-01-02	2	AR1	Fully meridional	None
3	2002-02-07	2002-02-08	2	AR1	Zonal-perpendicular	None
4	2002-02-24	2002-02-26	3	AR2	Concave	None
5	2002-12-11	2002-12-16	6	AR2	Zonal-like	4.7
6	2003-02-19	2003-02-21	3	AR1	Concave	4.9
7	2004-02-08	2004-02-10	3	AR1	Zonal-like	None
8	2005-01-03	2005-01-05	3	AR2	Zonal-like	5.8

4) The Background section currently provides separate introductions to the warming effects on the Antarctic Peninsula (AP), the impacts of ARs and föhn on the ice sheet, and the influence of AR-föhn events on AP temperatures. For a journal of Nature Communications' stature, this presentation is somewhat lengthy and could be significantly streamlined to present a more focused and concise narrative. And the Summary section primarily reiterates the main findings, resulting in repetition rather than synthesis. To meet the high standards of Nature Communications, this section should be evolved into a robust Discussion. It should delve into the implications of the results, address uncertainties, and outline unresolved questions that warrant future investigation

Thank you for this great suggestion. We have (1) trimmed the Background section and (2) rewritten the Discussion (formerly the Summary) to provide a more robust interpretation of the results.

Please kindly check Ln 1-102 & Ln 340-420.

5) The core novelty of the study lies in the classification of ARs and the subsequent analysis of the distinct atmospheric and radiative impacts of the four types. While valuable, this may not sufficiently constitute a major conceptual advance expected for a high-impact journal. To significantly strengthen the manuscript's contribution, I recommend expanding the analysis to investigate the downstream effects of these AR-föhn events on sea ice melt.

We appreciate this comment and agree with the reviewer that sea ice is an important subject for Antarctic AR studies. However, this topic is very complicated as sea ice can be impacted through multiple ways:

1. Dynamically: depending on the wind direction and sea ice distribution, AR can either increase sea ice thickness & decrease sea ice coverage or vice versa.
2. Thermodynamically: AR can significantly affect cloud formation thus radiative effects, with the potential for both warming and cooling impacts (Rowe et al. 2025a,b; Zou et al. 2023).
3. Oceanic forcings.
4. Impacts from precipitation (rainfall vs. snowfall).

A comprehensive response to these questions is better suited for a separate paper, which the lead author is currently preparing (experimental model studies targeting the impacts of SST and sea ice on AR development). As such, we expand the discussion of sea-ice implications and identify them as priorities for future work. We also highlight the potential application of AR shape along the broader Antarctic coastline, which is the subject of ongoing research.

Ln 395-409: *"Incorporating AR characteristics is essential for assessing surface impacts along the entire Antarctic coastline, as combined AR-föhn warming has been reported beyond the AP⁴⁴. This compound impact not only affects ice-shelf stability but may also trigger the disintegration and advection of sea ice in the downstream²⁰. Different AR configurations, characterized by distinct wind regimes and thermodynamic conditions⁴⁵, together with oceanic forcing⁴⁶, may be key to understanding sea-ice coverage and thickness over the AP. Under a warming and more moisture-rich future climate, stronger ARs may penetrate farther inland, facilitating greater moisture intrusion and more intense winds along the coast of the Antarctic continent⁴⁷. A higher probability of stronger ARs could consequently result in more frequent and severe surface warming events^{48,49} and sea ice retreat^{45,50}. Therefore, on individual ice shelves along the Antarctic coastline, those regional-scale atmospheric processes that are poorly resolved in global climate models will become increasingly important for ice shelf stability⁵¹ and sea ice conditions^{52,53}, ultimately influencing Antarctic ice loss and global sea-level rise. This further highlights the need to develop high-resolution regional coupled models that explicitly resolve interactions among Earth system components."*

Additional specific comments:

1) Figs 1 and 2: I suggest consolidating Figure 1 and Figure 2b into a single, more comprehensive figure. Meanwhile, Figure 2a could be moved to the supplementary information.

Thank you for pointing this out. We have merged this comment with a related comment from another reviewer. Accordingly, Fig. 1 has been updated, and Fig. 2a has been moved to the Supplementary Materials as Fig. S2.

“Fig. 1. PWRF model domains and overview of four AR shapes: a) D01 with a horizontal resolution of 30 km; b) D02 and D03 with resolutions of 6 km and 1.2 km, respectively; c) Classification of all events shown in Supplementary Fig. S1 into distinct AR shapes, which are zonal-perpendicular, zonal-like, convex, and concave. In panel b, the purple dashed line marks the AR detection zone; Spring Point, AWS 14, AWS 15 and AWS 17 are shown as red crosses, and the Escudero station as a purple dot. Topography in panels b-c is from the 6-km WRF D02 domain based on Reference Elevation Model of Antarctica (REMA). Larsen C denotes the Larsen C Ice Shelf.”

“Supplementary Figure S1. Dominant AR shape for each AR-föhn event impacting the AP during austral summers from 2001 to 2022, as well as corresponding dates and ranking on the Enhanced AR Scale¹. AR shapes of AR-family events are shown by dashed lines and a “F” after the date. If an AR-family event includes multiple shapes, the secondary shape is indicated by a dashed line. Also shown is the model topography (m)

from D02 at 6 km, based on the REMA dataset. LCIS in (b) refers to the Larsen C Ice Shelf. The dominant AR shape is manually defined based on the IVT field, following the AR axis at the peak of each event.”

2) Lines 185-186 and Fig.4: It would be helpful to see the spatial patterns of the latent heat flux and longwave radiation to better evaluate their roles in the surface energy budget and the resulting melting processes.

Thank you for this suggestion. Following the reviewer’s recommendation, we have updated both Figs 3 and S2. Related analysis has been updated too. Please check Ln 197-234:

“Fig. 3. Based on the k-means clustering shown in Fig. 2, composites of (a-d) the hourly mean rate of change of 2 m temperature ($\frac{dT_2}{dt}$; $\times 100$ $^{\circ}\text{C h}^{-1}$), (e-h) hourly average surface

downward shortwave radiation ($W m^{-2}$), (i-l) hourly average surface downward longwave flux ($W m^{-2}$) and (m-p) hourly average sensible heat flux ($W m^{-2}$) over the LCIS for zonal-perpendicular (a,e,i,m), zonal-like (b,f,j,n), concave (c,g,k,o) and convex (d,h,l,p) AR shapes. The diurnal cycle has been removed from the 2 m temperature, and values are multiplied by a factor of 100 for visualization.”

“Supplementary Figure S2. Based on the k-means clustering of Integrated Vapor Transport (IVT) shown in Fig. 2, composites of (a-d) hourly average latent heat flux ($W m^{-2}$), (e-h) hourly average net shortwave radiation ($W m^{-2}$) and (i-l) hourly average net longwave radiation ($W m^{-2}$) for zonal-perpendicular (a,e,i), zonal-like (b,f,j), convex (c,g,k) and concave (d,h,l) AR shapes.”

3) Lines 480-481: Please include a brief explanation of the physical significance of Fr values, specifically stating what $Fr > 1$ and $Fr < 1$ represent in the context of this study.

Thank you for this suggestion. After careful discussion among all authors, we decided to remove the Froude number calculation to avoid potential confusion, as the trajectory analysis and cross sections already clearly indicate whether low-level blocking occurs on the upwind side.

4) Lines 236-238 and Fig. 8: For the Fully Zonal AR type, the temporal evolution appears consistent with the patterns of IVT and downward longwave radiation. However, the SHF exhibits a lag of approximately 6 hours. To understand this discrepancy, it would be informative to examine the profile of diabatic heating.

Thank you for this suggestion. We believe this behavior arises from the strong spatial variability of SHF. Enhanced SHF is confined to regions near the mountain foothills or associated with föhn jet structures and is largely limited to the southern LCIS. If we look at the maximum SHF instead of median value, we can see a good match for both events, especially when IVT increases within the red box as shown in the figure below.

“Fig. 7. Box plots of six-hourly changes over the LCIS of (a-e) zonal-perpendicular and (f-j) convex AR events, occurring in December 2008 and February 2013 respectively, and based on PWRP simulations. (a,f) Sensible Heat Flux (SHF; $W m^{-2}$), (b,g) IVT ($kg m^{-1} s^{-1}$), (c,h) surface downward shortwave radiation (Downward SW; $W m^{-2}$), (d,i) surface downward longwave radiation (Downward LW; $W m^{-2}$), and (e,j) 2 m temperature (T_2 ; $^{\circ}C$) from 00Z 1 to 23Z 3 Dec and from 00Z 22 to 23Z 24 Feb, respectively. The colored solid line in each panel shows the median value, while the green dashed line in (a,f) indicates the maximum value.”

5) Lines 264-268 and Fig.8: For the convex AR case, the SHF does not exhibit a pronounced change and its temporal evolution does not align closely with the observed variations in 2-meter temperature.

As shown above, the median SHF (solid green) in Fig. 7 (originally Fig. 8) does not capture this localized enhancement, whereas the maximum SHF value exhibits a pronounced increase between 00 UTC on 22 February and 12 UTC on 23 February 2013.

We have added the following text:

Ln 293–295: *“Because SHF exhibited strong spatial variability and affected only limited areas, its median value remained relatively stable during the event.”*

6) Lines 283-297: These two sentences can be combined and made more concise.

Revised as

Ln 314–317: *“However, in both cases there was also a substantial contribution over the LCIS from the combined radiative and sensible heat flux term, which is negative and thus offset the isentropic drawdown contribution, with median values of $-8.5\text{ }^{\circ}\text{C}$ for the zonal-perpendicular event and $-14.9\text{ }^{\circ}\text{C}$ for the convex event.”*

7) Lines 287-289: Is SHF also a reduction?

SHF on average has a positive impact.

Revised as:

Ln 320: *“...resulting in a reduction in downward SW, despite the positive contributions from the SHF.”*

8) Table S1: suggested to add the AR type.

Thank you for this suggestion. The dormant AR shape and change of $T_{2\text{max}}$ over the LCIS are both added for each event in Supplementary Table S1.

**Please note that each event may exhibit multiple shapes; this table reflects only the dominant one.*

9) Table S2: Lack of "fully meridional"

Yes, the fully meridional case we selected is particularly interesting and therefore warranted a deeper investigation. Although this event is classified as an AR2 and IVT is detected over the Larsen C Ice Shelf, surface air temperature over Larsen C barely increases. This investigation is ongoing, and we plan to submit the related study to *Atmospheric Science Letters* in 2026.

10) Data: While the model performance is convincingly demonstrated for two representative cases, this validation would be strengthened by a more comprehensive evaluation. Could the authors please present statistical metrics (e.g., mean bias, correlation) that aggregate the results across all 37 simulated AR-föhn events?

Thank you for this suggestion. We have included additional evaluations based on available observations in Supplementary Table S4.

Spring Point is located on the upwind side of the Antarctic Peninsula, while AWS17 is situated on the leeward side. Both stations provide observations for more than 50% of the AR events period from 2001 to 2015.

The same modeling configuration was used in Zou et al. (2023), Gorodetskaya et al. (2023), Wille et al. (2024a,b), and Rowe et al. (2025a,b), all of which demonstrate reliable model performance.

Reviewer #2 (Remarks to the Author):

All of my concerns have been addressed by the authors.
I just have comments about a few grammatical errors in the sentences.

L109 : Table. S1 -> Table S1

L194, L212 : exhibites -> exhibits

L248 : approach -> approached

L338 : shows -> show

Thank you for pointing out these errors. They have been corrected in the revised manuscript.

Reviewer #3 (Remarks to the Author):

Thank you for your efforts in addressing all my initial comments. I am satisfied with most of the revisions. However, I still have a few remaining concerns and suggestions, as outlined below:

Thank you for all the suggestions. Please see the point-by-point responses below.

The definition and selection criteria for "AR-föhn events" could be more clearly articulated. I suggest providing a more detailed description of the identification process in the Methods section, including specific thresholds, time windows, and spatial extents, to enhance reproducibility and facilitate comparison with future studies.

Thank you for this suggestion. The requested information has now been added to the Methods section. We have also deposited example code together with the dataset to enhance reproducibility and facilitate comparisons with future studies.

Ln 514-515: *"For Antarctica, a ranking of AR1 to AR3 on the Enhanced AR Scale indicates moderate to strong events (IVT thresholds > 250 kg m⁻¹ s⁻¹)." - specific thresholds*

Ln 519-520: *We then computed the regional mean of this gridded dataset within the target area (60°S to 70°S, 45°W to 75°W; indicated by the purple dashed box in Fig. 1 and the red dashed box in Supplementary Fig. S1a). - spatial extents*

Ln 532-533: *"The maximum T2 over the LCIS increased by more than 3 °C for over 6 hours after removing the diurnal cycle..." - specific thresholds*

Ln 534- 535: *"The start of warming is defined as the onset of the maximum T2 increase, and the end as when maximum T2 decreases by more than 2 °C from its peak." - time windows*

While the cluster analysis successfully identified four distinct AR shapes, the authors also note that some shapes (e.g., zonal-perpendicular and concave) are not always

clearly distinguishable. I recommend including a discussion on this limitation and, if possible, exploring complementary classification methods or diagnostics to further validate the grouping.

Thank you for this suggestion. We have revised this and are actively looking for different methods to better classify AR shape (e.g., Self-organizing map). This will be a value aspect for our future investigation.

Ln 390– 391: “More advanced machine-learning techniques, such as self-organizing maps³⁹, may help better characterize AR shape and curvature.”

Regarding the AR scale classification, it would be helpful to explicitly highlight in the main text the limitations associated with AR1 and AR3 events, and to indicate whether future work will specifically address AR3 events in more detail.

Thank you for this suggestion. Some limitations and uninvestigated aspects remain for AR3, as several of these cases belong to AR-family events.

Revised as suggested:

Ln 379-380: “*Notably, three of the four AR-family events belong to AR3, warranting further investigation of this category and its compound impacts.*”

Please ensure that all abbreviations appearing in figures and tables are spelled out in full at their first occurrence, in accordance with standard journal guidelines.

Revised as suggested (e.g., Figs. 2 and 3). When space in the figure is limited, the full variable name is provided in the caption.